# Time Series, Vision, and Language:
# Exploring the Limits of Alignment in Contrastive Representation Spaces

**Pratham Yashwante** [1]   **Rose Yu** [1]

## Abstract

The Platonic Representation Hypothesis posits that learned representations from models trained on different modalities converge to a shared latent structure of the world. However, this hypothesis has largely been examined in vision and language, and it remains unclear whether time series participate in such convergence. We first examine this in a trimodal setting and find that independently pretrained time series, vision, and language encoders exhibit near-orthogonal geometry in the absence of explicit coupling. We then apply post-hoc alignment by training projection heads over frozen encoders using contrastive learning, and analyze the resulting representations with respect to geometry, scaling behavior, and dependence on information density and input modality characteristics. Our investigation reveals that overall alignment in contrastive representation spaces improves with model size, but this alignment is asymmetric: time series align more strongly with visual representations than with text, and images can act as effective intermediaries between time series and language. We further see that richer textual descriptions improve alignment only up to a threshold; training on denser captions does not lead to further improvement. Analogous effects are observed for visual representations. Our findings shed light on considerations for building multimodal systems involving non-conventional data modalities beyond vision and language. Code and data are available at https://github.com/Rose-STL-Lab/tvl-alignment.

[1]Department of Computer Science and Engineering, University of California San Diego, La Jolla, California, USA. Correspondence to: Pratham Yashwante <pyashwante@ucsd.edu>.

*Proceedings of the $43^{rd}$ International Conference on Machine Learning*, Seoul, South Korea. PMLR 306, 2026. Copyright 2026 by the author(s).

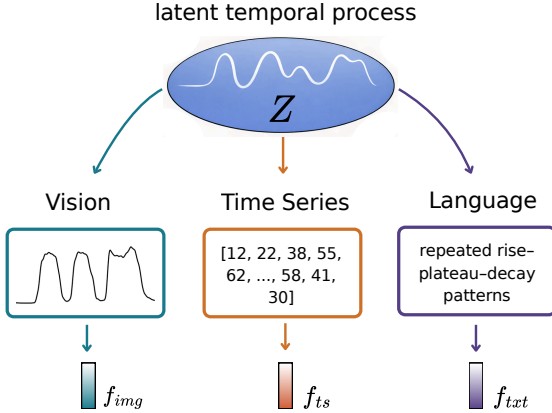

*Figure 1.* Trimodal projections of a shared temporal process. A latent process $Z$ gives rise to a numeric time series, a visual line plot, and a textual description, each representing the same signal in values, geometry, and language. Modality-specific encoders $f_{ts}$, $f_{img}$, and $f_{txt}$ map inputs into representation spaces.

## 1. Introduction

Neural network representations are increasingly converging. Across architectures and training objectives, deep models tend to organize inputs according to similar notions of similarity, an observation formalized as the Platonic Representation Hypothesis (PRH) (Huh et al., 2024). The hypothesis posits that representation learning models converge toward a shared statistical model of reality, shaped by the structure of the underlying data. Empirical evidence for this convergence is strongest for vision and language: as models scale, their induced similarity kernels become increasingly aligned. Joint vision–language (VL) models such as CLIP (Radford et al., 2021) further demonstrate that contrastive learning (CL) can produce coherent shared representations across perceptual and linguistic domains.

Time series pose a distinct challenge for multimodal alignment because their semantic structure is not directly observable from raw values. Images encode structure explicitly through spatial geometry, and text encodes semantics explicitly through symbolic tokens. In contrast, numeric time series express meaning primarily implicitly through temporal variation: properties such as trends, periodicity, or anomalies are not discrete tokens or visual features, but la-

tent properties that must be recovered from the signal. For example, while an image of a cat and the word "cat" explicitly denote the same concept, a temporal trend is neither an object nor a symbol, but a property that emerges only through computation over a numeric series. This mismatch raises a central question: *can time series achieve the same degree of representational alignment with vision and language?* Figure 1 illustrates the trimodal formulation studied in this work. Assuming that there exists a shared reality of a latent temporal process, we aim to understand the convergence behavior of learned representations from vision, language and time series.

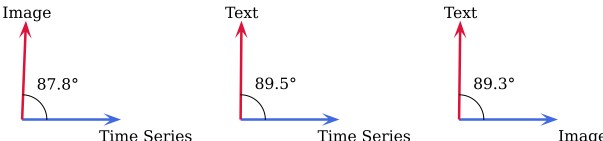

*Figure 2.* Mean angular deviation between pretrained cross-modal representations on CaTS shows little inherent alignment.

As a motivating observation, we first examine the representational geometry in the absence of explicit cross-modal coupling on the CaTS dataset (Zhou et al., 2026), which consists of triplets of numeric time series, corresponding line plots, and textual captions. We observe that independently pretrained cross-modal encoders exhibit near-orthogonal similarity across modalities (Figure 2). When external coupling via CL is applied, alignment behavior differs substantially across datasets: while image–text structure becomes coherent on Flickr (Plummer et al., 2015), the same objective gives more fragmented overlap on CaTS as shown in Figure 3, highlighting the unique challenges posed by time series. Additional analyses of uncoupled representational geometry across multimodal time series datasets are provided in Appendix A. These observations suggest that alignment may not emerge uniformly across modalities and datasets, even under identical contrastive objectives, thereby raising the question of how time series alignment differs from standard VL settings.

In this work, we conduct a systematic empirical study of trimodal alignment across time series, visual plots, and language, using CL as a controlled mechanism to probe representational compatibility across modality pairs. Throughout this work, the visual modality refers to rendered line plots of the time series, and the language modality refers to natural language descriptions; both reflect the dominant forms of supervision in real-world multimodal time series datasets. Our setup pairs pretrained encoders from each modality and trains projection heads to map representations into a shared space, following the CLIP paradigm. Our experiments span four datasets varying in domain and annotation richness, 34 encoder combinations across multiple model families and scales, and controlled ablations isolating the effects of

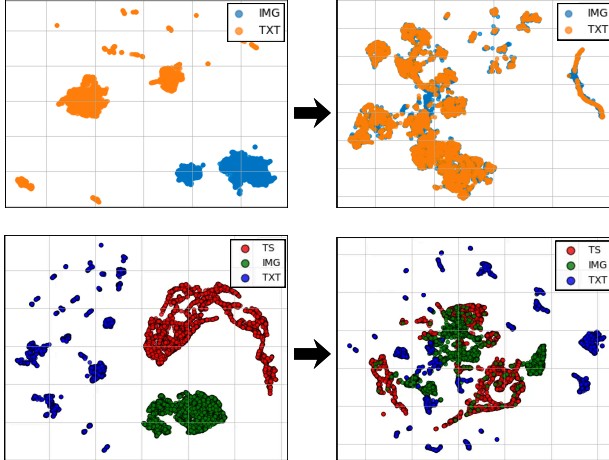

*Figure 3.* UMAP visualizations of representations after contrastive training on Flickr (top) and CaTS (bottom).

model capacity, optimization, and input characteristics.

Our contributions and empirical findings highlight several aspects of trimodal alignment involving time series:

1. **Trimodal Analysis:** We present the first systematic empirical study of trimodal alignment involving time series, images, and language, and discuss the roles of information density and semantic explicitness in governing cross-modal alignment.

2. **Asymmetric Convergence:** Trimodal alignment is asymmetric, with time series aligning more strongly with visual plots than with language. This gap persists even with detailed textual descriptions, providing evidence that multimodal convergence is uneven.

3. **Information Density Saturation:** Increasing information density improves alignment only up to a threshold, indicating that denser text by itself is insufficient to induce further cross-modal convergence.

4. **Grounding and Explicitness:** Alignment is shaped not only by model scale, but also by how explicitly semantics are grounded across modalities and how well representational formats are matched; modality pairs with more explicitly observable correspondences consistently align more strongly than those with implicit semantic links.

## 2. Related Work

**Contrastive learning.** CLIP (Radford et al., 2021) demonstrated that CL on large-scale image–text pairs produces representations with strong zero-shot transfer capabilities. The approach trains separate encoders for each modality and aligns their outputs using the InfoNCE loss (Oord et al., 2018), which maximizes agreement between matched pairs while minimizing agreement with mismatched pairs within a batch. Subsequent work has extended this paradigm to

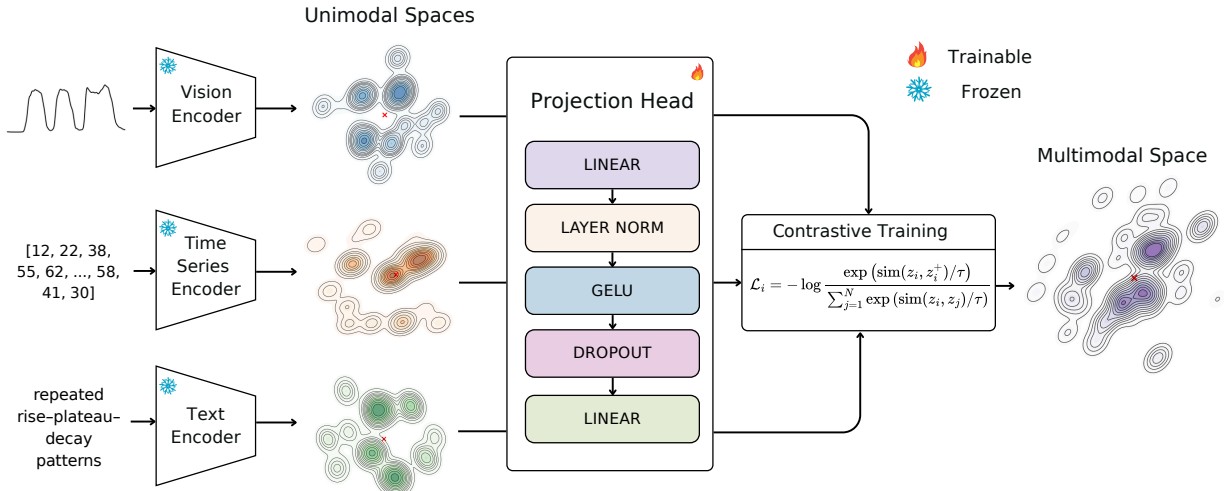

*Figure 4.* Trimodal contrastive alignment framework. Frozen pretrained encoders independently map time series, visual plots, and text into their respective unimodal representation spaces. Trainable projection heads transform these representations into a shared embedding space. Alignment is learned via a symmetric contrastive objective applied jointly across all modality pairs (TS–IMG, TS–TXT, IMG–TXT).

additional modalities including audio (Guzhov et al., 2022), video (Xu et al., 2021), and point clouds (Zhang et al., 2022).

**Time series models.** Recent work has introduced pretrained time series models for forecasting or masked prediction (Woo et al., 2024; Goswami et al., 2024; Ansari et al., 2024; Das et al., 2024). While these models demonstrate strong downstream performance, their representational alignment with other modalities remains underexplored. Prior work has also explored pairing time series with other modalities for specific applications. In healthcare, clinical notes have been aligned with physiological time series for patient outcome prediction (Baldenweg et al., 2024; Hayat et al., 2022). Contrastive learning has also been applied to time series–text settings, including MedCLIP (Wang et al., 2022b) and the recent TimesCLIP (Chen et al., 2025). Our work instead treats representational alignment as the main focus, studying when and under what conditions it emerges, with potential implications for downstream use.

**Semantic content and density in language.** The uniform information density hypothesis (Jaeger & Levy, 2006) proposes that speakers distribute information evenly across utterances to optimize communication efficiency. Subsequent work has operationalized this through surprisal-based metrics measuring how predictably information is distributed across tokens (Meister et al., 2021). We adapt these ideas to study how the amount and distribution of semantic content in text affects alignment quality.

Further discussion on representational alignment is provided in Appendix B, along with related work on multimodal electrocardiogram (ECG) representations in Appendix C.

## 3. Trimodal Alignment Framework

**Design Rationale.** As shown in Figure 4, our design adopts the core contrastive alignment mechanism similar to that used in CLIP, repurposing it as a controlled framework for analyzing trimodal representational compatibility.

We evaluate 34 trimodal configurations spanning 26 unique pretrained encoders, including 9 text, 9 vision, and 8 time series models across different scales. All configurations use identical hyperparameters for fair comparison. Encoder combinations appear in Appendix D, with experimental details and ablations in Appendix E. Each modality encoder is used as a frozen feature extractor, and its output is mapped into a shared embedding space via projection heads. All modalities use the same projection head architecture (Appendix E.7).

**Symmetric Contrastive Loss.** We align representations using a symmetric contrastive objective applied jointly across all modality pairs: time series–image (TS–IMG), time series–text (TS–TXT), and image–text (IMG–TXT). Given $\ell_2$-normalized embeddings $z_{\text{ts}}, z_{\text{img}}, z_{\text{txt}} \in \mathbb{R}^d$, we apply a bidirectional InfoNCE loss to each modality pair. For a modality pair $(x, y)$, the forward loss is defined as

$$\mathcal{L}_{x \to y} = -\frac{1}{N} \sum_{i=1}^{N} \log \frac{\exp\left(z_x^{(i)\top} z_y^{(i)}/\tau\right)}{\sum_{j=1}^{N} \exp\left(z_x^{(i)\top} z_y^{(j)}/\tau\right)}, \quad (1)$$

where $\tau$ is a temperature parameter. For example, the TS–IMG loss is defined as the average of the forward and reverse directions,

$$\mathcal{L}_{\text{ts-img}} = \tfrac{1}{2}\left(\mathcal{L}_{\text{ts} \to \text{img}} + \mathcal{L}_{\text{img} \to \text{ts}}\right). \quad (2)$$

Analogous losses are defined for TS–TXT and IMG–TXT (see Appendix E.3). The final training objective is the equally weighted sum of all three modality-pair losses:

$$\mathcal{L}_{\text{total}} = \mathcal{L}_{\text{ts-img}} + \mathcal{L}_{\text{ts-txt}} + \mathcal{L}_{\text{img-txt}}. \quad (3)$$

We also evaluate ablated variants that restrict the set of modality pairs in the loss, including bimodal TS–IMG and TS–TXT settings, as well as joint VL with time series (VL–TS; full details in Appendix E.4).

**Evaluation Metrics.** We evaluate alignment using metrics that capture global similarity, retrieval, and geometric consistency of cross-modal representations. All metrics are computed on held-out test sets using $\ell_2$-normalized embeddings. Implementation details and definitions are provided in Appendix F. For each modality pair, we calculate:

1. **Cosine similarity margin** defined as the difference between matched and mismatched pairs.
2. **Bidirectional cross-modal retrieval** using Recall@$k$ (R@1, R@5, R@10).
3. **Procrustes disparity** measures how well one modality can be aligned to another via an optimal orthogonal transformation.
4. **Centered Kernel Alignment (CKA)** measures non-linear representational similarity using RBF kernels.
5. **Mutual $k$NN** overlap measures agreement in local neighborhood structure.

We quantify the semantic richness of textual descriptions using *information density* (**ID**), defined as the total surprisal of a caption $x = (w_1, \ldots, w_T)$ under a language model:

$$\text{ID}(x) = T \cdot \bar{\ell}(x), \quad \bar{\ell}(x) = \frac{1}{T} \sum_{t=1}^{T} -\log p(w_t \mid w_{<t}), \quad (4)$$

where surprisal is computed using a frozen GPT-2 language model. Higher ID corresponds to captions that are longer and semantically more informative and less predictable; see Appendix H for full details. We use *semantic explicitness* to refer operationally to the degree to which a modality's encoding makes underlying temporal semantics and structure directly observable without requiring abstraction or inference (Appendix L for detailed discussion).

**Datasets.** We evaluate trimodal alignment using datasets that allow us to probe semantic explicitness, information density, and visual mediation. Because fully aligned trimodal time series datasets are rare, we combine human and synthetic data to construct modality-complete variants wherever necessary. Full details are provided in Appendix G.

We use **CaTS-Bench** (Zhou et al., 2026) as our primary dataset, as it natively provides aligned triplets of time se-ries, visual plots, and captions. The captions serve as *direct* textual representations, describing observable temporal structure such as trends, phases, events, and summary statistics. To study ID, we construct multiple caption variants derived from the same underlying time series, including progressively condensed test-set captions and an expanded high-ID variant. Condensed captions reduce the amount of explicitly described temporal content by limiting descriptions to a fixed number of salient phrases, while the high-ID variant exceeds the original captions by more than a factor of two in measured ID. We also evaluate on **TRUCE** (Jhamtani & Berg-Kirkpatrick, 2021), which contains short time series paired with concise, direct textual descriptions. Its short length enables explicit plot annotation of the time se-ries. Owing to its small size, it is used only for evaluation, with three visual variants per signal (generic, styled, and annotated) to assess visual robustness.

To study alignment under *indirect* textual supervision, we use **MIMIC** (Gow et al., 2023) and **PTB-XL** (Wagner et al., 2020), where text consists of diagnostic reports that do not explicitly describe waveform structure. MIMIC reports are in English, while PTB-XL reports are in German, allowing us to examine multilingual effects. For both datasets, we generate visual plots of ECG waveforms and construct train/test splits matched in size to CaTS. These datasets use longer series (5000 timesteps) than CaTS, enabling analysis of how temporal resolution affects multimodal alignment.

## 4. Results

**RQ1.** Do contrastive representation spaces converge uniformly as models scale?

Increasing model capacity is often associated with stronger representations (Jia et al., 2021; Radford et al., 2021; Hoffmann et al., 2022), but whether scale alone leads to convergence across time series, vision, and language remains an open question. We therefore use our trimodal framework to analyze alignment on CaTS across encoder scales, while holding the contrastive objective and parameters fixed.

Figure 5 reports alignment quality across modality pairs as a function of total model size where we summarize scaling trends using a linear model for interpretability and consistent comparison across modality pairs. We also fit LOESS (Cleveland & Devlin, 1988) and find that it support the same conclusions while further revealing saturation and variability effects at larger scales (Appendix E.8). Overall alignment improves with scale, but gains are highly uneven across modality pairs. TS–TXT alignment remains the weakest in absolute terms across all model sizes, reflecting the difficulty of directly aligning time series with captions, yet it also exhibits the strongest positive relationship with scale. In contrast, TS–IMG alignment achieves

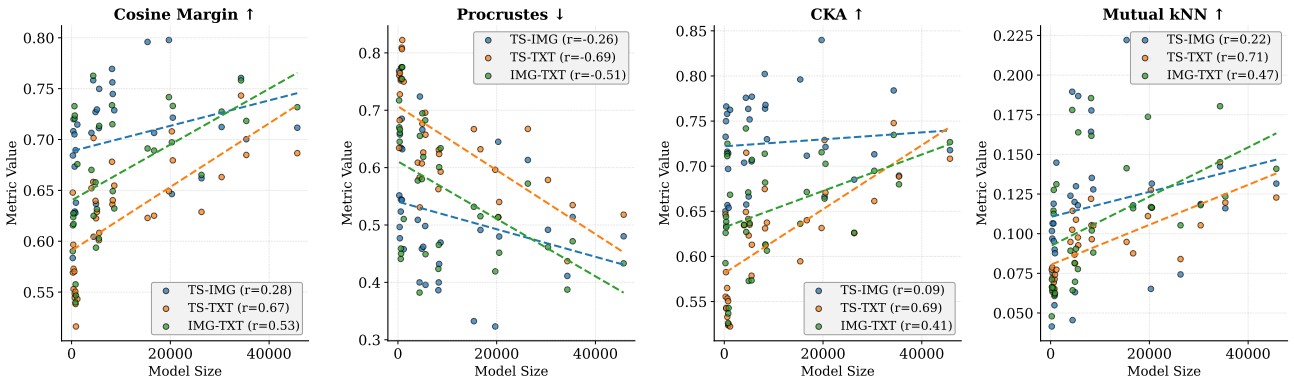

*Figure 5.* Scaling behavior of alignment across 34 trimodal configurations on CaTS as a function of total model size (in millions of parameters). Each subplot reports alignment quality for three modality pairs (TS–IMG, TS–TXT, IMG–TXT). Each point corresponds to a distinct encoder configuration. Dashed lines indicate linear trends with Pearson correlation coefficients reported in the legend.

high absolute performance even at smaller scales, but shows weaker scaling trends, suggesting earlier saturation driven by stronger shared visual grounding with temporal data.

Notably, global alignment across modalities is consistently strong, as reflected by cosine similarity and geometric metrics, while local neighborhood structure remains weak. Across all configurations, mutual $k$NN overlap stays low, indicating limited fine-grained correspondence. This highlights a clear dissociation between global and local alignment under CL: strong global geometric similarity does not imply robust neighborhood-level semantic correspondence.

Further, we observe a consistent asymmetry, with time series aligning more strongly with line plots than with text captions. This pattern reflects mismatched inductive biases and differences in semantic explicitness across modalities. Time series encode temporal properties implicitly such that a trend emerges only after computing differences over raw values. Visual plots externalize this structure: a trend becomes a visible slope, and a peak emerges as a distinct geometric pattern. Text abstracts further: a caption names "increasing trend" but provides no grounding in specific numerical or geometric form. This difference in how semantics and structure is exposed directly affects how easily representations can be matched in a shared embedding space. As a result, alignment is strongest between embeddings of modalities that expose semantics in comparable formats (TS–IMG), while implicit-to-abstract alignment (TS–TXT) remains weaker. Detailed discussion in Appendix L.

We also verify that this asymmetry is not an artifact of the deterministic relationship between time series and their visual plots. Temporally scrambled plots, which preserve visual format while destroying temporal semantics, cause cosine similarity to drop by $79.5\%$ and Procrustes distance to increase by $210.7\%$, consistently across four DINOv2 configurations, confirming that alignment reflects shared temporal semantics rather than determinism (Appendix I).

An interesting direction is whether alternative textual representations such as code, which encode more structured context, can reduce the TS–TXT alignment gap. We find that code improves alignment for encoders pretrained on diverse text corpora, suggesting a potential intermediate level of explicitness in our setting: TS–IMG > TS–TXT (code) > TS–TXT (NL). However, gains depend on the encoder's ability to parse structured syntax, and the fundamental asymmetry persists (Appendix L). Also, comparing synthetic and human-written CaTS captions, we observe only modest shifts, showing that learned embeddings with the trimodal framework generalize well to natural linguistic variability (Appendix K).

Finally, we also find that trimodal alignment is sensitive to both optimization quality and encoder capacity. Larger batch sizes and higher-capacity projection heads consistently improve alignment which confirms the importance of effective contrastive optimization. Also, scaling the time series encoder provides substantial gains for both TS–IMG and TS–TXT alignment. This identifies temporal representations as a key factor in trimodal alignment, with improvements to the time series encoder being important for strengthening otherwise weak modality pairs such as TS–TXT. Additional analyses and ablations are provided in Appendix E.6 (optimization), E.7 (projection head design), and J (time series encoder capacity).

> **RQ2.** Do pretrained VL models change alignment?

Jointly pretrained VL models exhibit strong intrinsic alignment between images and text (Radford et al., 2021; Li et al., 2022; Zhai et al., 2023), raising the question of whether such coupling affects trimodal alignment. In particular, it is unclear whether strong IMG–TXT alignment in trimodal systems needs to be learned jointly, or can instead be inherited from existing VL pretraining. To isolate this effect, we conduct this comparison on CaTS, contrasting the full

trimodal setting with configurations where we pair a pre-trained VL model with a time series encoder. We train both settings using the same contrastive objective, allowing differences in alignment to be attributed to representational priors.

As shown in Figure 6, we see that VL–TS configurations achieve strong IMG–TXT alignment even at relatively small scales. This suggests the tightly coupled image–text geometry inherited from VL pretraining, which enables robust alignment without relying on increased model capacity. In contrast, trimodal encoders must jointly learn structure across all three modalities, and therefore depend more heavily on scale to compensate for weaker intrinsic coupling.

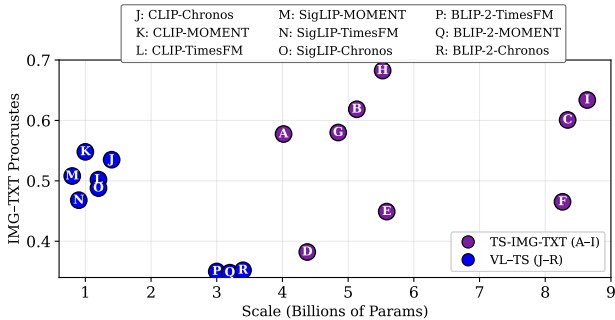

*Figure 6.* IMG–TXT alignment across model scales for VL–TS on CaTS. A–I correspond to select configurations (Appendix D.2).

> **RQ3.** How does information density in text affect cross-modal alignment?

Because textual descriptions vary widely in how explicitly they encode semantic structure about an underlying time series, we study how the *information density* of text affects the strength of cross-modal alignment. We vary textual semantic richness by constructing multiple caption variants for each sample with different ID. Specifically, we use large language models (LLMs) (GPT-4o-mini and LLaMA-3.2-90B) to compress the original captions to a fixed number of salient semantic phrases, while keeping the underlying data, model architectures, and contrastive setup fixed. We observe that alignment quality improves consistently with increasing ID across all metrics, as shown in Figure 7. Using the original CaTS captions with three progressively condensed variants, we find that as captions become denser and more information-rich, Procrustes disparity decreases, indicating tighter shared geometry, while $k$NN overlap increases, reflecting stronger neighborhood-level semantic consistency.

Low-information captions provide insufficient semantic signal, leading to weakly structured or partially collapsed embeddings and high variability across model configurations. At very low ID, embeddings cluster tightly with limited neighborhood differentiation, indicating that sparse seman-

tic content constrains the formation of meaningful relational structure. In contrast, higher-information captions encode richer attributes and contextual relationships, enabling more coherent cross-modal geometry. Larger models benefit most in this regime, but also degrade more sharply when captions are short and under-specified, reflecting a stronger dependence on explicit semantic content. These results show that the amount of explicitly encoded semantic information plays a key role in multimodal alignment when text is involved.

Since alignment improves as captions become denser, we test whether further increasing caption ID during training provides additional gains. We test whether training on captions with more than double the original CaTS ID improves alignment. We find that despite this substantial increase in semantic content, alignment metrics change only marginally across all modality pairs as shown in Table 1. This indicates that once text reach sufficient semantic richness, further increases in ID do not meaningfully improve alignment.

*Table 1.* Effect of doubling text ID on alignment. We compare original captions (train ID = 417.2) and high-information captions (train ID = 870.2). Δ denotes High ID − OG ID. Both models use same samples with their respective ID-specific caption sets. Scores are averaged across configurations listed in Appendix D.2.

| Pair | Cosine Margin ↑ | | | Procrustes ↓ | | | Mutual kNN ↑ | | |
|---|---|---|---|---|---|---|---|---|---|
| | OG | High | Δ | OG | High | Δ | OG | High | Δ |
| TS–IMG | 0.72 | 0.72 | 0.00 | 0.47 | 0.47 | 0.00 | 0.13 | 0.13 | 0.00 |
| TS–TXT | 0.57 | 0.56 | -0.01 | 0.74 | 0.74 | 0.00 | 0.08 | 0.07 | -0.01 |
| IMG–TXT | 0.64 | 0.64 | 0.00 | 0.60 | 0.60 | 0.00 | 0.11 | 0.10 | -0.01 |

A similar effect appears when evaluating CaTS-trained models on TRUCE for TS–TXT alignment. TRUCE captions are short, direct textual descriptions with ID comparable to our low-ID (2 phrase) caption variants (ID ≈ 74.5 vs. 69.6) and similar length (12.5 vs. 10.2 words on average), and give comparable levels of TS–TXT alignment. This mirrors the behavior observed for CaTS captions in the low-ID regime and provides additional evidence that alignment is strongly governed by the amount and structural organization of semantic content in the text modality.

> **RQ4.** What happens when text does not directly describe the time series signal?

In many real-world settings, text associated with time series provides high-level interpretations rather than direct descriptions of signal structure. We therefore ask whether alignment persists when language is only indirectly related to the underlying signal. To answer this, we evaluate trimodal alignment using the same contrastive framework under *indirect* textual supervision using ECG datasets. Specifically, we compare CaTS with MIMIC, where clinical reports only implicitly relate to waveform dynamics.

We observe distinct alignment scaling behavior between

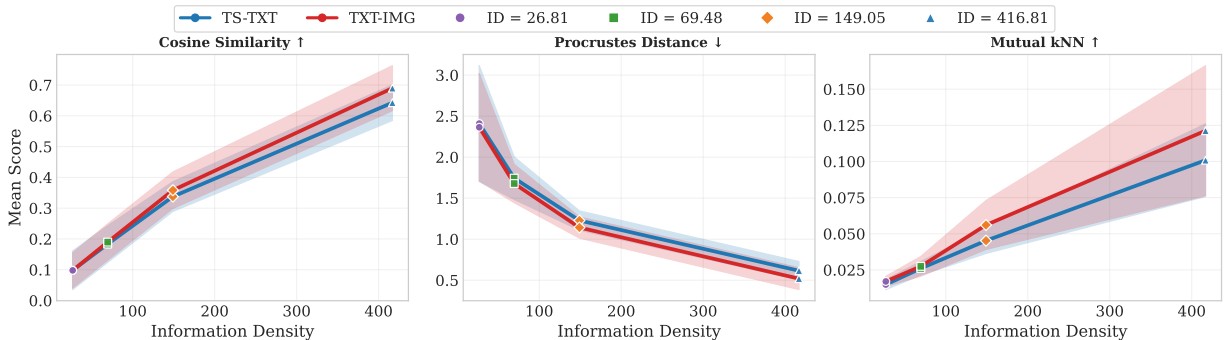

*Figure 7.* Effect of text ID on alignment across increasing levels of text density. Markers indicate distinct caption regimes with increasing semantic richness. Scores are averaged across 9 representative model configurations (see Appendix D.2).

CaTS and MIMIC across different trimodal configurations (Figure 8). Although increasing model capacity improves alignment across all modality pairs in both datasets, systematic differences persist depending on the nature of textual supervision. Note that CaTS captions exhibit substantially higher textual density (test ID = 416.81), whereas MIMIC reports have much lower ID (test ID = 149.48), and this disparity is directly reflected in cross-modal alignment strength.

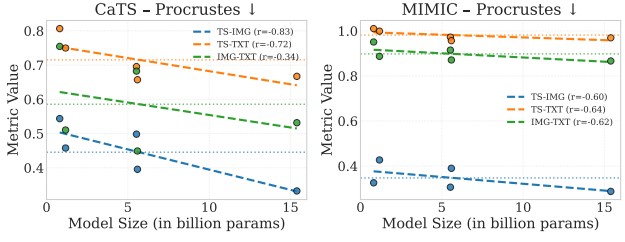

*Figure 8.* Scaling behavior of alignment on CaTS and MIMIC across 5 representative trimodal configurations (see Appendix D.2). We report Procrustes disparity as a function of model size.

Alignment involving text is consistently weaker on MIMIC, particularly for TS–TXT and IMG–TXT pairs. Even at comparable model scales, MIMIC exhibits lower cosine similarity and higher Procrustes disparity, indicating poorer global and geometric alignment when language provides only indirect semantic grounding. This gap is most pronounced for TS–TXT which shows that clinical language provides substantially less explicit signal for aligning numeric time series than direct descriptive captions. Scaling partially mitigates this effect: larger models improve TS–TXT alignment on both datasets, but gains are substantially weaker under indirect supervision, with alignment remaining consistently worse on MIMIC across scales, suggesting a limitation associated with semantic explicitness.

In contrast, TS–IMG alignment is often stronger on MIMIC despite weaker text alignment. This reflects the longer and more structured ECG signals in MIMIC, which provide richer numeric structure for alignment with visual plots.

We also see that stronger TS–IMG alignment also yields large gains in cross-modal retrieval (Table 2), confirming that improved geometric alignment translates into retrieval performance even when textual grounding is weak.

*Table 2.* Cross-modal retrieval performance on MIMIC and CaTS for TS↔IMG retrieval. Results are macro-averaged over TS→IMG and IMG→TS and reported as percentages. **Model acronyms:** Dv2/Dv3 = DINOv2/DINOv3, SL = SigLIP2, Q = Qwen, Mo = MOMENT, C = Chronos; B/L denote Base/Large.

| Model Configuration | Retrieval Performance (%) | | | | | |
|---|---|---|---|---|---|---|
| | R@1 | | R@5 | | R@10 | |
| | CaTS | MIMIC | CaTS | MIMIC | CaTS | MIMIC |
| Dv2-B + Q-0.6B + Mo-B | 1.61 | 31.31 | 6.13 | 61.84 | 10.75 | 73.62 |
| Dv2-L + Q-4B + Mo-L | 2.25 | 21.36 | 8.65 | 49.48 | 14.71 | 62.21 |
| Dv3-7B + Q-8B + Mo-L | 6.19 | 23.70 | 18.35 | 51.80 | 27.12 | 65.13 |
| SL2-B + Q-0.6B + C-B | 3.96 | 20.33 | 12.64 | 45.79 | 19.85 | 58.69 |
| SL-L + Q-4B + C-L | 6.95 | 29.78 | 19.84 | 58.45 | 28.65 | 69.90 |

**RQ5.** How sensitive is alignment to language shifts?

Thus far, we have examined how alignment depends on explicitness and density within a single language. In practice, however, textual supervision can also vary linguistically. To isolate the effect of language shift, we compare alignment between MIMIC and PTB-XL, which both pair ECG time series with clinical reports from the same domain but differ in report language (English vs. German).

Figure 9 shows that alignment is consistently weaker on PTB-XL across all modality pairs. This degradation is pronounced for pairs involving text, indicating that linguistic shift further weakens semantic coupling even when the underlying time series distribution is comparable. All trends observed in earlier experiments persist: TS–TXT remains the most challenging pair, increased scale provides partial compensation, and TS–IMG alignment remains relatively robust for longer more structured time series. These results suggest that cross-modal alignment is sensitive not only to semantic explicitness and text ID, but also to how well the

language of textual supervision aligns with the inductive biases of pretrained language encoders.

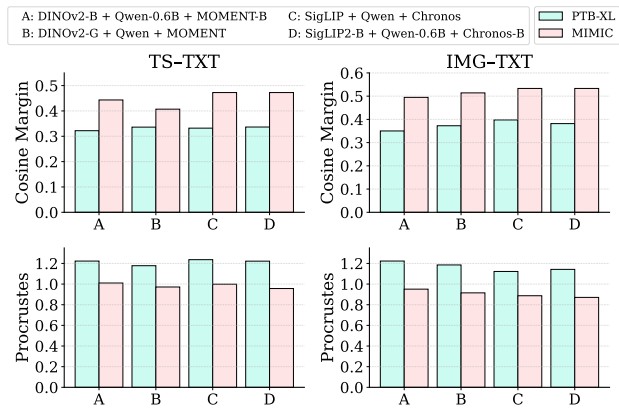

*Figure 9.* Comparison of multimodal alignment between MIMIC (English reports) and PTB-XL (German reports). English reports consistently achieve higher alignment across all modality pairs.

We also construct MIMIC-DE by translating MIMIC English reports to German using GPT-4o-mini to isolate the effect of language shift from dataset-level differences, keeping all other factors fixed. We observe that alignment degrades progressively from MIMIC-EN to MIMIC-DE to PTB-XL-GE, which suggests that linguistic shift weakens alignment even when signal distribution and report structure remain constant (Appendix C.1, Table 3).

**RQ6.** Do richer visual inputs help alignment?

While earlier results show that textual semantic explicitness and information density strongly affect alignment, it remains unclear whether similar effects arise in the visual modality. To test whether increasing visual semantic content improves alignment, we compare TS–IMG alignment across plot variants from the TRUCE dataset with progressively richer visual annotations (Appendix G.2), holding the underlying time series and training setup fixed.

Figure 11 shows that increasing visual semantic richness leads to consistent improvements in TS–IMG alignment across configurations. Annotated plots achieve higher alignment than generic or stylistically varied variants, indicating that richer and denser visual inputs provide stronger semantic anchors for aligning time series with images. Consistent with earlier trends, increasing model capacity (compare D and E) further improves TS–IMG alignment which indicates that density and scale act as complementary factors.

In Appendix I, we further provide a comprehensive set of controls showing that TS–IMG alignment is driven by temporal semantics and structure. These controls include robustness to stylistic perturbations of line plots, where alignment remains stable and improves with richer styling. We also

evaluate alternative visual encodings such as recurrence plots, where strong alignment persists despite structural shifts in representation. Finally, under temporally scrambled inputs that preserve visual appearance but destroy temporal order, the alignment degrades sharply. All of these results indicate that alignment is sensitive to preserved temporal structure and semantic grounding.

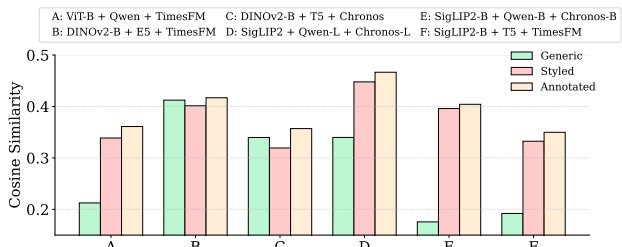

*Figure 11.* TS–IMG alignment across TRUCE plot variants with increasing visual input richness. Annotated achieves highest scores.

**RQ7.** Does trimodality help weakly aligned pairs?

Given the persistent asymmetry between TS–IMG and TS–TXT alignment, we ask whether introducing a third modality can facilitate alignment for otherwise weak modality pairs. To answer this, we measure alignment changes when one modality (IMG or TXT) is removed, comparing bimodal and trimodal contrastive learning on CaTS (Figure 10).

For the weakest modality pair, TS–TXT, introducing the image modality produces a consistent upward shift across alignment metrics which suggests that visual representations provide intermediate semantic structure that neither time series nor text capture in isolation. This improvement reflects not only better global geometry but also in neighborhood structure, which suggests that images help organize weakly coupled representations. Recent work consistently supports this: (Ruan et al., 2024) and (Xue et al., 2023) show that adding images improves text-to-3D alignment, while (Cicchetti et al., 2026) and (Zhong et al., 2025) demonstrate that a third modality (audio or vision) resolves ambiguities in video-text and time series-text pairs. These findings confirm that trimodality benefits weakly aligned pairs by supplying intermediate semantic structure.

We further test whether image embeddings lie geometrically between time series and text embeddings in the shared space, providing geometric evidence for this intermediary role. We find that 73.4% of TS–IMG–TXT triplets in configuration 12 (Table 4) place the image embedding closer to both endpoints than they are to each other. We also measure triangle-chain error, defined as the mean relative residual when the image is used to bridge TS and TXT. The mean triangle-chain error drops from 0.90 for DINOv2 configurations to 0.34 for SigLIP2 configurations, indicat-

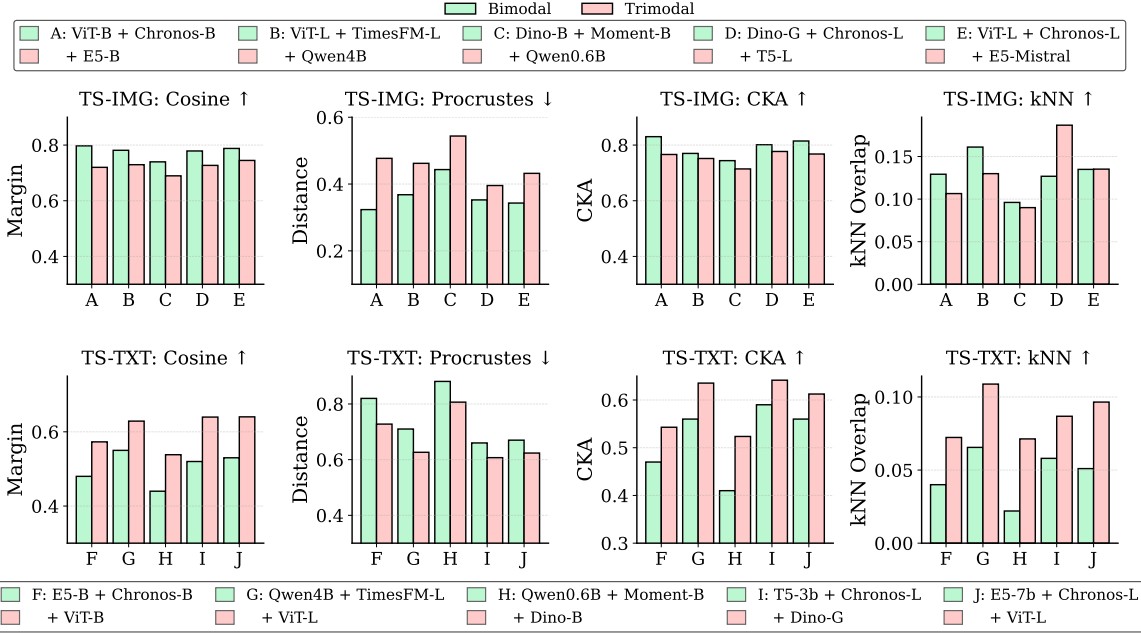

*Figure 10.* Role of trimodality as a semantic bridge. We compare bimodal and trimodal CL for TS–IMG (top row) and TS–TXT (bottom row). Introducing the image modality consistently improves TS–TXT alignment.

ing that VL pretraining shapes image representations to be more semantically compatible with both modalities simultaneously, further strengthening their role as intermediaries (Appendix L).

For a strong pair (TS–IMG), introducing a third modality often degrades performance, suggesting that it adds optimization complexity without providing additional semantic grounding. This shows that trimodality is most effective when it introduces missing semantics rather than redundant supervision.

## 5. Discussion

We first find that independent time series, vision, and language encoders produce near-orthogonal representations, indicating that multimodal convergence with time series does not arise without explicit coupling in our setting. With contrastive alignment, convergence remains asymmetric: vision and language align readily, while time series lag, with the weakest performance in TS–TXT. This asymmetry persists across scales and reflects differences in semantics, where time series are implicit, text symbolic, and visual plots explicitly expose temporal structure. As a result, images can act as intermediaries for weak modality pairs.

We further observe that semantic explicitness constrains recoverable structure: increasing information density improves alignment only up to a saturation point, indicating representational mismatch rather than scale or supervision

as an important bottleneck. While scaling and optimization improve weaker pairs, architectural coupling and pretraining objectives remain influential, as evidenced by jointly pretrained VL models.

Practically, images can serve as effective bridges, structured text outperforms longer captions, and scaling time series encoders shows consistent gains, highlighting representation quality as a key limitation. Alignment also correlates with improved cross-modal retrieval performance (Table 13, Appendix N.2).

Our analysis is limited to univariate time series, and it remains unclear whether the observed behavior extends to multivariate signals. CaTS captions are synthetic and may not fully reflect natural language diversity, while clinical datasets rely on indirect supervision and domain-specific language. We use frozen encoders with learned projection heads for controlled comparison, which isolates representational compatibility but excludes end-to-end fine-tuning and modality-specific adaptation.

Extending this analysis to sensor-based time series (e.g. Ego4D (Grauman et al., 2022)) would help test generalization and connect with methods such as ImageBind (Girdhar et al., 2023) and IMU2CLIP (Moon et al., 2023). Another direction is to study how alignment relates to downstream performance, which is limited by the scarcity of datasets that jointly contain time series, vision, and language. Developing such benchmarks and methods would clarify how representational alignment translates to practical utility.

## Acknowledgement

This work was supported in part by NSF Grants #2205093, #2146343, #2134274, CDC-RFA-FT-23-0069, the U.S. Army Research Office under Army-ECASE award W911NF-07-R-0003-03, the U.S. Department Of Energy, Office of Science, IARPA HAYSTAC Program, and DARPA YFA. We also thank the anonymous reviewers for their helpful comments and suggestions.

## Impact Statement

This paper aims to advance the empirical understanding of multimodal representation learning involving time series, visual, and language modalities. The work focuses on analyzing alignment behavior under controlled experimental settings, with the goal of clarifying how different modalities interact in contrastive representation spaces. While the study is not intended as a deployment-facing system and does not introduce models trained for direct decision-making in real-world applications, the questions it addresses are relevant to domains such as scientific analysis and healthcare, where multimodal data are common. We do not foresee immediate negative societal impacts arising from this work. By improving understanding of the conditions under which multimodal representations align or fail to align, this work aims to inform more robust evaluation practices and support the responsible development of future multimodal systems.

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

# A. Learned Convergence Without External Coupling

Before doing post-hoc alignment, we first understand whether representations learned independently within each modality already exhibit convergent structure, as suggested by PRH for vision and language models, as an initial motivation. We extend this diagnostic to a trimodal setting involving time series, vision, and language, and evaluate representational geometry in the absence of any external coupling. For each dataset for their test sets, we extract embeddings from independently pretrained encoders for time series, images, and text. No projection heads are trained and no contrastive or other alignment objectives are applied. We then compare geometry directly using mean angular deviation (MAD) and cosine similarity.

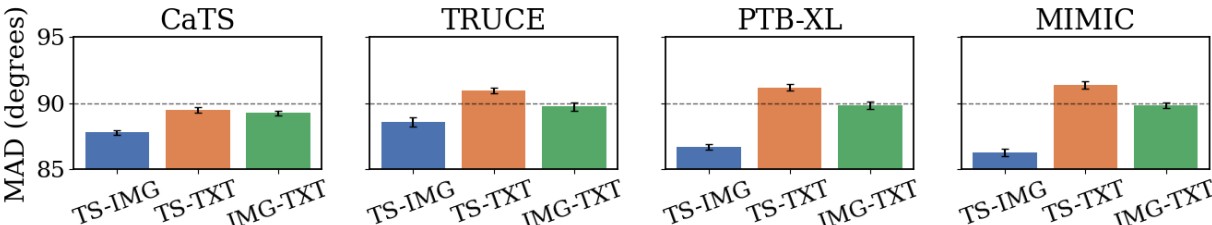

*Figure 12.* MAD between modality pairs across datasets and model configurations. Values remain near $90°$ for all pairs, indicating near-orthogonal geometry and a lack of inherent cross-modal convergence in independently pretrained encoders.

Figure 12 reports MAD between modality pairs across 9 representative trimodal configurations which we used in VL-TS experiment (A–I) (see Appendix D.2) and four datasets. Across all datasets and modality pairs, MAD values remain close to $90°$, indicating near-orthogonal geometry. This behavior is consistent across domains, including cases where representations are deterministic renderings of the underlying signal (CaTS, TRUCE) and cases where text is indirectly related to the signal (PTB-XL, MIMIC). These results indicate that independently pretrained encoders do not exhibit inherent geometric convergence across these datasets used. Figure 13 visualizes cross-modal cosine similarity matrices computed over matched samples. Off-diagonal similarities remain uniformly close to zero across all datasets, with no discernible separation between matched and mismatched pairs. This also confirms that the lack of alignment is not merely a matter of scale but reflects the absence of latent cross-modal correspondence in the embedding spaces. Notably, TS–IMG similarity also remains close to zero in this uncoupled setting, although it is consistently higher than other pairs.

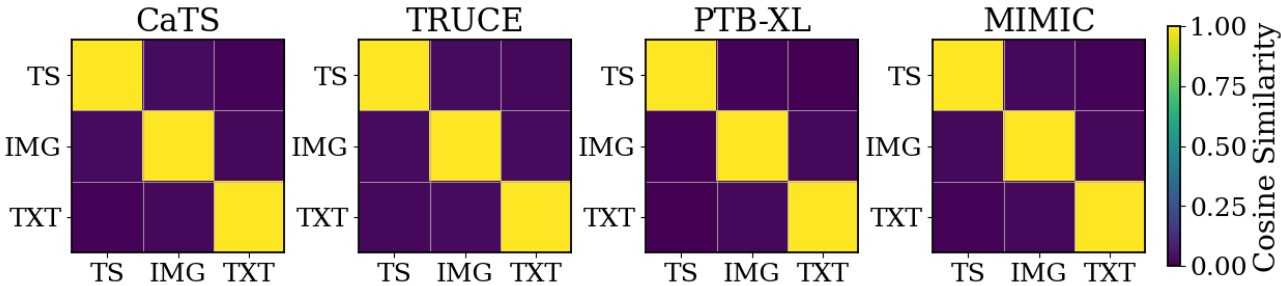

*Figure 13.* Cross-modal cosine similarity matrices for matched samples across datasets. Off-diagonal similarities remain near zero with no separation between matched and mismatched pairs, confirming the absence of latent alignment without explicit coupling.

These diagnostics establish a clear baseline: independently pretrained encoders for time series, vision, and language do not spontaneously converge to a shared representational geometry for different modality projections of time series data on the datasets studied here. This motivates the contrastive alignment experiments in the paper, where we introduce trainable projection heads to study how post-hoc alignment is induced.

# B. Continued Related Work

A long-standing theme across machine learning and computational neuroscience is that learned representations can be meaningfully compared across models, layers, training runs, and even across fundamentally different modalities. Early work in neuroscience formalized this idea through representational similarity analysis (RSA), which compares the geometry of activation patterns via representational (dis)similarity matrices rather than attempting neuron-to-neuron correspondence

(Kriegeskorte et al., 2008). This perspective has strongly influenced modern deep learning research, where alignment is typically operationalized as the extent to which two embedding spaces share structure up to simple transformations.

Though a central challenge in measuring representational similarity is that different models may implement equivalent functions using different bases, permutations, or scalings which makes direct coordinate-wise comparison ill-defined. Canonical correlation analysis (CCA) and its variants address this by comparing subspaces rather than individual units, motivating widely used tools such as SVCCA (Raghu et al., 2017) and projection-weighted CCA (PWCCA) (Morcos et al., 2018). CKA further refines this approach by providing a similarity index that is stable across random initializations and closely connected to CCA while importantly remaining computationally tractable (Kornblith et al., 2019). Another common metric of alignment is Procrustes analysis, which measures how well two representation sets can be matched via an optimal orthogonal transformation, with classical roots in the orthogonal Procrustes problem (Schönemann, 1966). More recent work has begun to systematically benchmark and compare these similarity measures across architectures, datasets, and modalities, demonstrating that different metrics emphasize different invariances (Klabunde et al., 2025; Tjandrasuwita et al., 2025).

Beyond measurement, lots of works explicitly seeks to induce alignment through training objectives. Early multiview learning methods such as deep canonical correlation analysis (DCCA) learn nonlinear transformations of two views whose outputs are maximally correlated (Andrew et al., 2013), with extensions to more than two views enabling multiway alignment through DGCCA (Benton et al., 2019). In modern models, cross modal alignment is most commonly induced via contrastive learning objectives. CLIP (Radford et al., 2021) and ALIGN (Jia et al., 2021) demonstrated that large-scale image–text contrastive training can produce shared embedding spaces with strong zero-shot transfer properties, while distillation-based methods such as ALBEF improve robustness and alignment quality under noisy supervision (Li et al., 2021). Theoretical and empirical analyses of contrastive learning characterize these objectives as balancing alignment between positive pairs and uniformity of the representation distribution (Wang & Isola, 2020). While large-scale contrastive training can induce shared representation spaces across modalities, less is known about the geometry, asymmetry, and limits of such alignment particularly when modalities have fundamentally different inductive biases such as time series.

Contrastive alignment has been extended beyond image–text to audio–text (Elizalde et al., 2023), image–text–audio (Guzhov et al., 2022), video–text (Xu et al., 2021), time series-text (Buiting et al., 2025), and 3D–text (Hadgi et al., 2025). More unified frameworks align multiple modalities within a shared embedding space, including VATT, which jointly learns from raw video, audio, and text (Akbari et al., 2021), and ImageBind (Girdhar et al., 2023), extends a VL backbone to modalities via image-paired training. A growing line of work also explores multimodal time series models that adapt large language models or unify modalities in shared spaces, including Time-LLM-style methods (Jin et al., 2024; Zhou et al., 2023; Gruver et al., 2023) and vision-language–based frameworks such as TimeVLM (Zhong et al., 2025).

## C. ECG–Text Alignment and Textual Specificity

A substantial recent literature has explored aligning ECG waveforms with clinical text using contrastive and fine-grained supervision for downstream tasks, including MERL (Liu et al., 2024), MELP (Wang et al., 2025), FG-CLEP (Li et al., 2025), and ECG-Chat (Zhao et al., 2025). These works report strong zero-shot and linear-probe performance on downstream ECG tasks, often substantially exceeding unimodal baselines. While our study does not propose a new ECG–text alignment architecture, our findings offer a representation-centric perspective that provides insight into why these methods are effective.

A key distinction across this literature lies in how textual supervision is constructed and used. Methods such as MELP and FG-CLEP explicitly introduce *fine-grained or structured supervision*, including beat-level, rhythm-level, or entity-aware alignment, thereby creating localized correspondences between waveform segments and textual elements. In contrast, MERL relies on global contrastive alignment during training but achieves strong performance through carefully engineered, knowledge-augmented prompting at inference time, which increases the specificity and clinical grounding of textual queries. Despite differing mechanisms, these approaches can be viewed as sharing a common goal which is reducing the semantic gap between low-level waveform structure and high-level clinical language. Our results are consistent with this interpretation. We find that alignment involving indirect or high-level text is consistently weaker than alignment with visual projections of the signal. In ECG datasets, this manifests as strong TS–IMG alignment and retrieval performance, while TS–TXT and IMG–TXT alignment remains comparatively weaker, though stronger than in synthetic settings such as CaTS.

From this perspective, prior ECG–text methods can be understood as introducing mechanisms that effectively convert indirect text into more *direct* supervision. Fine-grained losses (e.g., beat- or token-level alignment), entity extraction, validated waveform descriptors, or knowledge-enhanced prompting all increase semantic explicitness and local grounding,

thereby addressing the representational mismatch we identify. Our analysis provides a representation-level interpretation of the empirical gains reported by these methods. Importantly, our work complements these approaches by isolating textual specificity as a key factor governing alignment quality, and we provide a controlled analysis that helps clarify why ECG–text alignment is fundamentally more challenging than ECG–image alignment, and why additional structure or supervision is often required. This view offers guidance for future multimodal ECG systems, suggesting that improvements are more likely to arise from increasing semantic explicitness and local grounding than from scaling contrastive objectives alone.

### C.1. Isolating Language Shift from Dataset Differences

We construct MIMIC-DE by translating MIMIC English reports to German using GPT-4o-mini to test whether the alignment gap between MIMIC and PTB-XL reflects language shift specifically, keeping all other factors fixed. Manual inspection of a random subset of the constructed dataset confirmed preservation of clinical terminology and diagnostic content. As shown in Table 3, TS–TXT alignment degrades progressively from MIMIC-EN to MIMIC-DE to PTBXL-GE, confirming that linguistic shift independently weakens alignment. The residual gap between MIMIC-DE and PTB-XL-GE further suggests that dataset-level factors contribute additionally beyond language alone.

*Table 3.* Effect of language shift on TS–TXT alignment (DINOv2-B + Qwen-0.6B + MOMENT-B).

| Setting | Cosine Margin ↑ | Procrustes ↓ |
|---|---|---|
| MIMIC-EN | 0.44 | 1.01 |
| MIMIC-DE (translated) | 0.38 | 1.09 |
| PTB-XL-GE | 0.32 | 1.22 |

## D. Encoders and Trimodal Combinations

**Models used.** We evaluate a broad and diverse set of pretrained models across vision, language, and time series modalities, covering multiple architectural paradigms, training objectives, and model scales.

1. **Vision encoders.** We use nine vision encoders spanning model sizes from 86M to 18B parameters. Self-supervised models include DINOv2 (Oquab et al., 2024) (Base, 86.6M; Giant, 1.14B) and DINOv3 ViT (7B) (Siméoni et al., 2025). Supervised models include ViT (Dosovitskiy et al., 2021) (Base, 86.4M; Huge, 632M). Vision–language pretrained models include SigLIP (Zhai et al., 2023) (Large, 878M), SigLIP2 (Tschannen et al., 2025) (Base, 375M), and EVA-CLIP (Sun et al., 2023) (8B, 18B).

2. **Text encoders.** We evaluate nine text encoders with scales ranging from 60M to 27B parameters. These include T5 (Raffel et al., 2020) (Small, 60.5M; 3B), E5-Base-v2 (109M) and E5-Mistral (7B) (Wang et al., 2022a; 2024), Qwen3-Embedding (Yang et al., 2025; Zhang et al., 2025) (596M, 4B, 8B), Mistral-Nemo-Base (12B) (Mistral AI & NVIDIA, 2024), and Gemma-3 (27B) (Gemma Team et al., 2025).

3. **Time series encoders.** We use eight pretrained time series encoders with model sizes ranging from under 100M to over 700M parameters. These include TimesFM (Das et al., 2024) (200M, 500M), Chronos (Ansari et al., 2024) (201M, 709M), MOMENT (Goswami et al., 2024) (125M, 385M), and Moirai (Woo et al., 2024) (91M, 311M).

### D.1. Representation Extraction

We adopt a consistent and controlled representation extraction protocol across text, vision, and time series models to ensure fair comparisons across modalities and families. All backbone models are kept frozen and used strictly as feature extractors.

1. **Text representations.** For text models (Qwen, Gemma, and Mistral), we extract token-level hidden states from the final transformer layer and apply mean pooling over valid tokens using the attention mask, yielding a single fixed-dimensional sentence embedding per sample. For embedding-style models (e.g., Qwen Embedding), we use the last-token or sequence-end representation following the model's recommended pooling strategy.

2. **Vision representations.** Vision representations are obtained by encoding images through pretrained vision backbones (e.g., EVA-CLIP, DINOv3), using either the model's native image embedding head or the [CLS] token from the final layer, depending on the architecture.

3. **Time series representations.** For time series encoders, we extract sequence-level embeddings by mean pooling over

temporal hidden states returned by the encoder.

In all cases, the resulting modality-specific embeddings are projected into a shared latent space of fixed dimension using trainable projection heads, while the backbone encoders remain frozen.

*Table 4.* Triplet model combinations used in our experiments (Grouped by Size Bounds)

| # | Vision Encoder | Text Encoder | Time Series Encoder |
|---|---|---|---|
| | *Lower Bound Models* | | |
| 1 | DINOv2-Base (86.6M) | T5-Small (60.5M) | Moirai (91M) |
| 2 | ViT-Base (86.4M) | T5-Small (60.5M) | MOMENT (125M) |
| 3 | DINOv2-Base (86.6M) | T5-Small (60.5M) | Chronos (201M) |
| 4 | DINOv2-Base (86.6M) | E5-Base-v2 (109M) | TimesFM (200M) |
| 5 | ViT-Base (86.4M) | E5-Base-v2 (109M) | Chronos (201M) |
| 6 | SigLIP2-Base (375M) | E5-Base-v2 (109M) | Moirai (91M) |
| 7 | SigLIP2-Base (375M) | E5-Base-v2 (109M) | MOMENT (125M) |
| 8 | SigLIP2-Base (375M) | T5-Small (60.5M) | TimesFM (200M) |
| 9 | ViT-Base (86.4M) | Qwen3-Embedding (596M) | Moirai (91M) |
| 10 | DINOv2-Base (86.6M) | Qwen3-Embedding (596M) | MOMENT (125M) |
| 11 | ViT-Base (86.4M) | Qwen3-Embedding (596M) | TimesFM (200M) |
| 12 | SigLIP2-Base (375M) | Qwen3-Embedding (596M) | Chronos (201M) |
| | *Middle Bound Models* | | |
| 13 | ViT-Huge (632M) | T5 (3B) | MOMENT (385M) |
| 14 | SigLIP (878M) | T5 (3B) | TimesFM (500M) |
| 15 | DINOv2-Giant (1.14B) | T5 (3B) | Moirai (311M) |
| 16 | DINOv2-Giant (1.14B) | T5 (3B) | Chronos (709M) |
| 17 | ViT-Huge (632M) | Qwen3-Embedding (4B) | Moirai (311M) |
| 18 | ViT-Huge (632M) | Qwen3-Embedding (4B) | TimesFM (500M) |
| 19 | DINOv2-Giant (1.14B) | Qwen3-Embedding (4B) | MOMENT (385M) |
| 20 | SigLIP (878M) | Qwen3-Embedding (4B) | Chronos (709M) |
| 21 | SigLIP (878M) | E5-Mistral (7B) | Moirai (311M) |
| 22 | SigLIP (878M) | E5-Mistral (7B) | MOMENT (385M) |
| 23 | ViT-Huge (632M) | E5-Mistral (7B) | Chronos (709M) |
| 24 | DINOv2-Giant (1.14B) | E5-Mistral (7B) | TimesFM (500M) |
| | *Upper Bound Models* | | |
| 25 | DINOv3 ViT (7B) | Qwen3-Embedding (8B) | MOMENT (385M) |
| 26 | EVA-CLIP (8B) | Qwen3-Embedding (8B) | Chronos (709M) |
| 27 | DINOv3 ViT (7B) | Mistral-Nemo-Base (12B) | Chronos (709M) |
| 28 | EVA-CLIP (8B) | Mistral-Nemo-Base (12B) | Moirai (311M) |
| 29 | EVA-CLIP (8B) | Mistral-Nemo-Base (12B) | TimesFM (500M) |
| 30 | EVA-CLIP (18B) | Qwen3-Embedding (8B) | Moirai (311M) |
| 31 | EVA-CLIP (18B) | Mistral-Nemo-Base (12B) | MOMENT (385M) |
| 32 | DINOv3 ViT (7B) | Gemma-3 (27B) | Moirai (311M) |
| 33 | EVA-CLIP (8B) | Gemma-3 (27B) | MOMENT (385M) |
| 34 | EVA-CLIP (18B) | Gemma-3 (27B) | Chronos (709M) |

### D.2. Representative Configuration Subsets

1. **Information density analysis (Figure 7).** Results are averaged over 9 representative configurations corresponding to rows {31, 29, 32, 34, 14, 22, 3, 9, 2} in Table 4.
2. **VL–TS scaling analysis (Figure 6).** Configurations labeled A–I in Figure 6 correspond to rows {A: 13, B: 18, C: 23, D: 14, E: 20, F: 22, G: 16, H: 19, I: 24} in Table 4.

3. **Scaling under indirect textual supervision (Figure 8).** Results are reported over 5 representative configurations {10, 19, 25, 12, 20} in Table 4.

4. **High-ID saturation analysis (Table 1).** Results are averaged over configurations {10, 20, 12, 5} from Table 4.

## E. Experimental Details, Ablations, Loss Functions

In this section, we describe the experimental protocol used to train and evaluate our models, including optimization settings, batching and precision choices, data handling, and the contrastive objective. We additionally present targeted ablations to assess the sensitivity of alignment geometry to optimization hyperparameters, training duration, and projection head design.

### E.1. Optimization, Batching and Precision

Models are trained for 50 epochs using AdamW with learning rate $1 \times 10^{-4}$ and weight decay $1 \times 10^{-5}$. A cosine decay schedule with linear warmup (2% of total steps) is applied. A fixed temperature of 0.2 is used throughout. Gradient clipping with maximum norm 1.0 is used to stabilize training. Early stopping with patience 5 is applied uniformly. All experiments use an effective batch size of 256, obtained via gradient accumulation when necessary. Mixed-precision (bfloat16) is used for very large encoders. All hyperparameters are held constant across datasets and model combinations.

### E.2. Data Handling

Time series keep their original length and no truncation is applied. Images are resized to 384×384 pixels; Text inputs use a maximum length of 512 tokens; all samples fall within this limit, so full text is retained across all variants.

### E.3. Loss

Let a minibatch consist of $N$ aligned triplets $\{(z_{\text{ts}}^{(i)}, z_{\text{img}}^{(i)}, z_{\text{txt}}^{(i)})\}_{i=1}^N$, where all embeddings are $\ell_2$-normalized and lie in $\mathbb{R}^d$. Similarity is measured using the dot product $s(a,b) = a^\top b$ and scaled by a temperature parameter $\tau > 0$.

For a generic modality pair $(x, y)$, the forward contrastive loss $x \to y$ is defined as

$$\mathcal{L}_{x \to y} = -\frac{1}{N} \sum_{i=1}^N \log \frac{\exp\left(s(z_x^{(i)}, z_y^{(i)})/\tau\right)}{\sum_{j=1}^N \exp\left(s(z_x^{(i)}, z_y^{(j)})/\tau\right)} \tag{5}$$

The reverse direction $y \to x$ is defined analogously:

$$\mathcal{L}_{y \to x} = -\frac{1}{N} \sum_{i=1}^N \log \frac{\exp\left(s(z_y^{(i)}, z_x^{(i)})/\tau\right)}{\sum_{j=1}^N \exp\left(s(z_y^{(i)}, z_x^{(j)})/\tau\right)}. \tag{6}$$

The symmetric bidirectional loss for modality pair $(x, y)$ is then

$$\mathcal{L}_{x\text{-}y} = \frac{1}{2}\left(\mathcal{L}_{x \to y} + \mathcal{L}_{y \to x}\right). \tag{7}$$

In our trimodal setting, we compute losses for all three modality pairs: time series–image (TS–IMG), time series–text (TS–TXT), and image–text (IMG–TXT). The final training objective is the equally weighted sum:

$$\mathcal{L}_{\text{total}} = \mathcal{L}_{\text{ts-img}} + \mathcal{L}_{\text{ts-txt}} + \mathcal{L}_{\text{img-txt}}. \tag{8}$$

In all cases, negative examples are implicitly provided by other samples within the same minibatch, i.e., each non-matching pair $(z_x^{(i)}, z_y^{(j)})$ for $j \neq i$ is treated as a negative. No external negative mining or memory bank is used.

**Equal Weighting of Modality Pairs in the Loss.** We apply equal weights to all modality pairs in the trimodal contrastive objective. This choice reflects our goal of studying intrinsic representational compatibility rather than optimizing for a

downstream task. Reweighting individual losses would introduce assumptions about the relative importance of different modality relationships, which would directly shape the geometry learned during training. In our setting, we treat each modality as a peer projection of a shared latent process, and no modality pair is favored *a priori*. There is therefore no principled basis for assigning greater weight to one interaction over another.

For example, increasing the weight of TS–TXT would implicitly assume that linguistic alignment is more semantically important, while downweighting TS–IMG could assume that visual renderings are easier or partially redundant. Such choices would be arbitrary and closely tied to the hypothesis under study. Using equal weights provides the most neutral formulation, allowing cross-modal structure to emerge from the representations themselves rather than being imposed through optimization choices. Notably, the observed asymmetry between TS–IMG and TS–TXT alignment persists across model scales, encoder families, and text richness despite symmetric treatment in the loss. This suggests that the asymmetry reflects underlying representational compatibility. Exploring adaptive or modality-specific weighting is a natural direction for task-oriented settings for multimodal time series tasks, but is outside the scope of our goal.

### E.4. Hierarchical Training Objective for Joint VL-TS combinations

We train all models in this setting using a hierarchical contrastive objective that aligns time series representations with joint VL semantics, while preserving internal VL consistency.

Let $z_{ts} \in \mathbb{R}^d$ denote the projected embedding of a time series input, and let $z_{vl} \in \mathbb{R}^d$ denote the corresponding joint VL embedding obtained from the multimodal backbone. We optimize the symmetric InfoNCE loss for $\mathcal{L}_{ts \leftrightarrow vl}$. The reverse direction $\mathcal{L}_{vl \rightarrow ts}$ is also applied analogously. This objective encourages each time series embedding to align most strongly with its corresponding multimodal semantic representation, while repelling mismatched pairs within the batch. To preserve semantic coherence within the joint VL space, we use an auxiliary contrastive loss between vision-only and text-only embeddings to observe IMG–TXT alignment.

$$\mathcal{L}_{v \leftrightarrow t} = \frac{1}{2}\left(\mathcal{L}_{v \rightarrow t} + \mathcal{L}_{t \rightarrow v}\right), \tag{9}$$

where $\mathcal{L}_{v \rightarrow t}$ and $\mathcal{L}_{t \rightarrow v}$ follow the same InfoNCE formulation.

### E.5. Compute

Training cost varies substantially with model scale and the number of trainable parameters. For lower-bound configurations, each triplet combination required approximately 6–10 hours of training, while middle- and upper-bound configurations required between 1 and 3 days per run. All experiments were executed using 8 NVIDIA A100 GPUs on a shared high-performance computing node with sufficient CPU and memory resources to support large-scale multimodal training. These runtimes reflect the cost of training projection heads with frozen encoders.

### E.6. Setting Ablations

We assess the sensitivity of trimodal alignment to contrastive optimization by comparing two configurations: *b64* (batch size 64, temperature 0.7, 512-dimensional projection) and *b256* (batch size 256, temperature 0.2, 1024-dimensional projection). We find that the *b256* configuration consistently gives stronger alignment across modality pairs. These gains reflect the combined effects of larger negative sets, sharper contrastive separation, and higher-capacity projection heads. Differences are most pronounced in global geometric metrics: Procrustes disparity remains low for *b256* (approximately 0.45) but degrades substantially under *b64* (above 0.70).

**Epoch Ablations.** We study the effect of training duration by performing an epoch ablation on a representative configuration, training the projection head for up to 100 epochs while keeping all encoders frozen. As summarized in Table 5, metrics improve rapidly during early training and largely stabilize by 40–50 epochs. Extending training beyond this point yields no consistent improvements and in some cases leads to slight degradation in alignment which shows diminishing returns once cross-modal correspondences are established. This suggests that prolonged optimization of the projection head can introduce geometric drift without improving semantic alignment. Based on these observations, we adopt a fixed training budget of 50 epochs for all experiments which helps us to balance computational efficiency with stable and robust alignment.

*Table 5.* Epoch ablation on `vit_h14 + t5 + moment-l`. Metrics are averaged across all modality pairs.

| Epochs | Cosine Margin ↑ | Procrustes ↓ | CKA ↑ | Mutual KNN ↑ |
|--------|-----------------|--------------|-------|--------------|
| 50 | **0.676** | **0.565** | **0.658** | 0.103 |
| 100 | 0.671 | 0.577 | 0.637 | **0.104** |

### E.7. Projection Head Ablations

We study the effect of projection head design on alignment by comparing a simple baseline projection head with a more expressive variant, which we use as our default throughout the paper. The baseline head consists of a shallow two-layer MLP with a ReLU nonlinearity, similar to very early contrastive learning setups in the field. Given an encoder output $x$, the improved projection head is defined as:

$$
\begin{aligned}
h(x) &= \mathrm{LayerNorm}(W_1 x + b_1), \\
u(x) &= \mathrm{Dropout}(\mathrm{GELU}(h(x))), \\
\mathrm{Proj}(x) &= W_2 u(x) + b_2.
\end{aligned}
\tag{10}
$$

where $W_1 \in \mathbb{R}^{m \times d_{\mathrm{in}}}$, $W_2 \in \mathbb{R}^{d \times m}$, $m = \max(3d, 768)$, and $d = 1024$ is the shared embedding dimension. The improved projection head increases capacity by introducing an expanded hidden layer, layer normalization, GELU activation, and dropout which provides more flexible feature re-mapping while keeping all encoders frozen. As shown in Figure 14, the improved projection head gives substantial gains in cosine similarity margins across all modality pairs. At the same time, Procrustes disparity is uniformly reduced, indicating that the learned embedding spaces are not only closer in a pointwise sense but also better aligned geometrically. These improvements hold across different families and combinations. Importantly, while the improved head strengthens alignment, the relative ordering between modality pairs remains stable, indicating that projection head design improves alignment quality without altering the underlying cross-modal asymmetries.

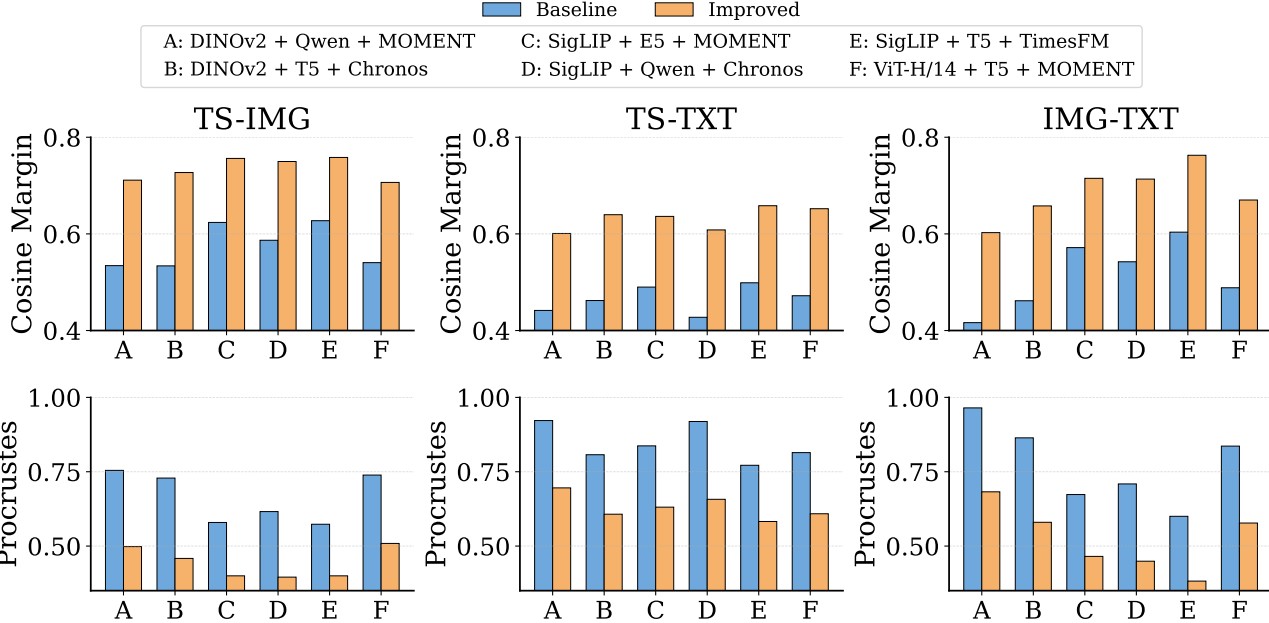

*Figure 14.* Effect of projection head design on trimodal alignment. Comparison of baseline and improved projection heads across six model configurations. The top row shows cosine similarity margins (higher is better) for TS–IMG, TS–TXT, and IMG–TXT alignment, while the bottom row reports Procrustes disparity (lower is better).

We additionally experimented with deeper projection head variants that further increase depth and nonlinearity beyond the improved design (e.g., additional hidden layers and normalization), but did not observe consistent gains over the reported

configuration. In most cases, alignment metrics saturated or showed marginal improvements at the cost of increased instability and sensitivity to hyperparameters. As a result, we adopt the improved projection head as a principled trade-off between capacity and robustness. We leave a more thorough exploration of end-to-end encoder fine-tuning or parameter-efficient adaptation to future work. While such approaches may further improve cross-modal alignment for our trimodal setting, they substantially increase computational cost and complicate fair comparison across heterogeneous encoder families, which is a central goal of this study.

We additionally performed a preliminary parameter-efficient finetuning experiment using LoRA on one representative configuration (DINOv2-B + Qwen-0.6B + and MOMENT-B). While finetuning improved alignment across all modality pairs, the overall asymmetry remained unchanged, with TS–IMG alignment consistently stronger than TS–TXT. Specifically, cosine similarity increased from 0.6895 to 0.7248 for TS–IMG and from 0.5383 to 0.5939 for TS–TXT, while Procrustes disparity decreased from 0.5437 to 0.5006 and from 0.8066 to 0.7598 respectively. These results suggest that the observed asymmetry reflects intrinsic representational differences. These preliminary findings indicate that our conclusions are robust to end-to-end adaptation. This observation is also consistent with the projection head ablations above, where increasing projection head capacity improved alignment without altering the relative ordering between modality pairs.

### E.8. Alternative Scaling Trend Fits

In the main paper, Figure 5 summarizes scaling trends using linear regression for interpretability and consistent comparison across modality pairs. We additionally fit LOESS curves to the same data. Figure 15 shows the LOESS fits across the alignment metrics. The LOESS fits are consistent with the conclusions drawn from the linear trends: overall alignment improves with scale, TS–TXT exhibits the strongest positive relationship with scale, and TS–IMG achieves higher absolute alignment even at smaller scales. Beyond this, LOESS reveals two additional effects. First, TS–IMG alignment saturates early and plateaus at larger scales, particularly visible in Cosine and Procrustes, while TS–TXT continues to improve. Second, we also observe non-monotonic behavior at small model sizes, reflecting higher variability among lower-bound configurations rather than a complete dip in alignment quality.

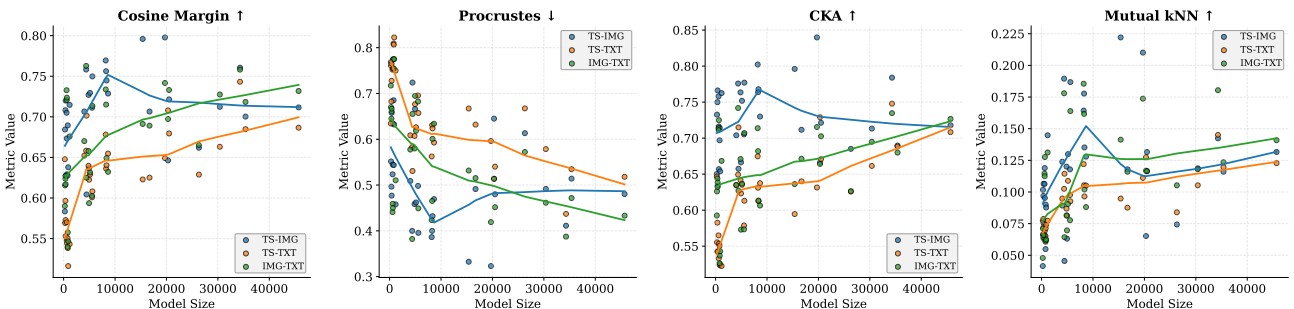

*Figure 15.* LOESS fits to scaling trends across 34 trimodal configurations on CaTS.

## F. Metrics

We quantify cross-modal alignment using complementary metrics that capture global geometric correspondence, non-linear representational similarity, and local neighborhood consistency. All metrics are computed independently for each modality pair (TS–IMG, TS–TXT, IMG–TXT) using normalized embeddings. To ensure computational efficiency and stability, all geometric metrics are computed on a randomly sampled subset of at most 2000 paired samples when datasets exceed this size, using a fixed seed (42). Cosine is done on full test set.

Let $\mathbf{X} = \{x_i\}_{i=1}^N$ and $\mathbf{Y} = \{y_i\}_{i=1}^N$ denote paired embeddings from two modalities, where each embedding is $\ell_2$-normalized.

### F.1. Cosine Similarity

Cosine similarity serves as a global semantic alignment metric. For paired samples, we compute the full similarity matrix

$$S = \mathbf{X}\mathbf{Y}^\top, \tag{11}$$

where $S_{ij} = \langle x_i, y_j \rangle$. We report three statistics:

$$\text{Matched} = \frac{1}{N}\sum_{i=1}^{N} S_{ii}, \quad \text{Mismatched} = \frac{1}{N(N-1)}\sum_{i \neq j} S_{ij}, \quad \text{Margin} = \frac{1}{N}\sum_{i=1}^{N}\left(S_{ii} - \frac{1}{N-1}\sum_{j \neq i} S_{ij}\right). \tag{12}$$

The margin reflects how well matched cross-modal pairs are separated from mismatched pairs. Higher values indicate stronger alignment.

### F.2. Procrustes Disparity (Global Geometry)

We compute the normalized Procrustes disparity between embedding spaces. After centering both modalities,

$$\tilde{\mathbf{X}} = \mathbf{X} - \frac{1}{N}\sum_{i} x_i, \tag{13}$$

$$\tilde{\mathbf{Y}} = \mathbf{Y} - \frac{1}{N}\sum_{i} y_i, \tag{14}$$

we solve the orthogonal Procrustes problem

$$\min_{\mathbf{R} \in \mathbb{R}^{d \times d}, \; \mathbf{R}^\top \mathbf{R} = \mathbf{I}} \|\tilde{\mathbf{X}}\mathbf{R} - \tilde{\mathbf{Y}}\|_F^2. \tag{15}$$

The optimal rotation $\mathbf{R}$ is obtained via the Kabsch solution using singular value decomposition with explicit reflection correction. We report the normalized disparity

$$\text{Disparity} = \frac{\|\tilde{\mathbf{X}}\mathbf{R} - \tilde{\mathbf{Y}}\|_F^2}{\|\tilde{\mathbf{Y}}\|_F^2}. \tag{16}$$

Lower values indicate stronger global geometric alignment.

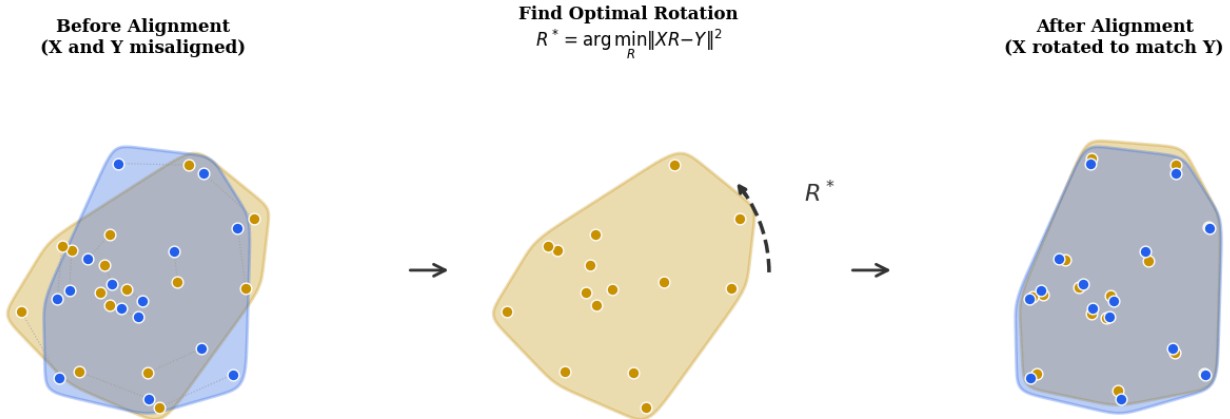

*Figure 16.* Illustration of Procrustes alignment. Embeddings from one modality are rotated to best match the other under an orthogonal transformation. Normalized residual error measures global geometric mismatch.

### F.3. CKA (Non-linear Similarity)

To capture non-linear representational similarity, we compute CKA using an RBF kernel. Given embeddings $\mathbf{X}$ and $\mathbf{Y}$, we compute

$$K_{ij} = \exp\left(-\gamma \|x_i - x_j\|^2\right), \tag{17}$$

$$L_{ij} = \exp\left(-\gamma \|y_i - y_j\|^2\right). \tag{18}$$

The bandwidth $\gamma$ is selected using the median heuristic over pairwise distances from both modalities. Kernels are centered using

$$\tilde{K} = HKH, \quad \tilde{L} = HLH, \tag{19}$$

where $H = I - \frac{1}{N}\mathbf{1}\mathbf{1}^\top$ and where $\mathbf{1} \in \mathbb{R}^N$ denotes the all-ones vector. CKA is then computed as

$$\text{CKA}(K, L) = \frac{\langle \tilde{K}, \tilde{L} \rangle_F}{\|\tilde{K}\|_F \|\tilde{L}\|_F}. \tag{20}$$

Values closer to $1$ indicate highly similar non-linear representational structure.

### F.4. Mutual $k$-Nearest Neighbors (Local Structure)

To evaluate local neighborhood consistency, we compute mutual $k$-nearest neighbor overlap. For each embedding $x_i$, let $\mathcal{N}_k^X(i)$ denote its $k$ nearest neighbors in $\mathbf{X}$ under cosine distance, excluding itself. Similarly define $\mathcal{N}_k^Y(i)$ for $\mathbf{Y}$. The mutual $k$NN score is

$$\text{Mutual-}k\text{NN} = \frac{1}{Nk} \sum_{i=1}^{N} \left| \mathcal{N}_k^X(i) \cap \mathcal{N}_k^Y(i) \right|. \tag{21}$$

This metric captures local neighborhood agreement across modalities. Higher values indicate stronger preservation of neighborhood structure. We use $k = 5$ in all computations as want to preserve the results on local structure and increasing it more would turn it into a global geometry metric.

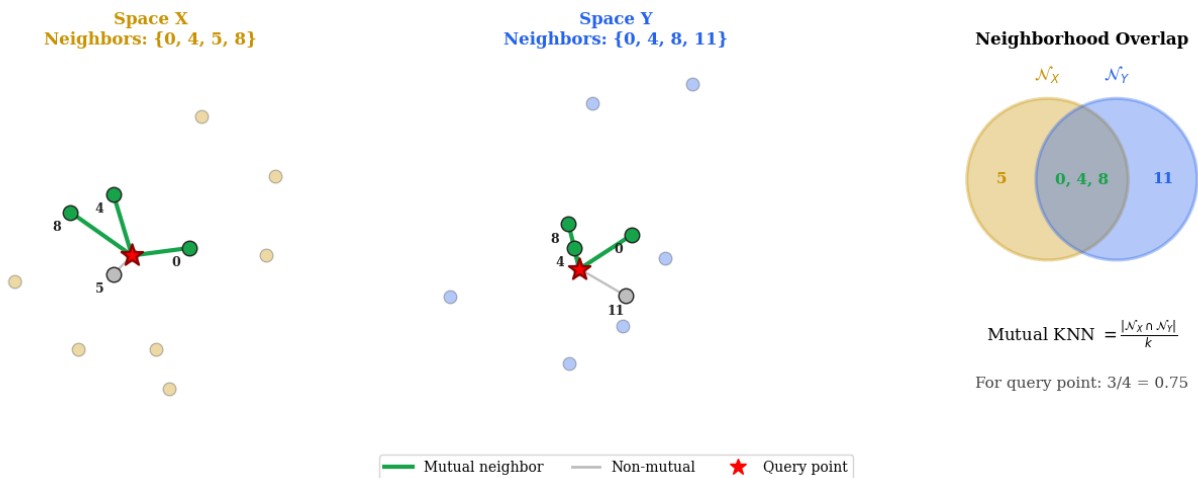

*Figure 17.* Illustration of Mutual $k$NN alignment. Mutual $k$NN measures local neighborhood agreement across modalities. Points that share neighbors in both spaces contribute to higher scores.

### F.5. Cross-modal retrieval.

For each modality pair (TS–IMG, TS–TXT, IMG–TXT), we evaluate cross-modal retrieval by treating embeddings from one modality as queries and ranking all embeddings from the paired modality over the full evaluation set using cosine similarity in the shared embedding space. For a dataset of $N$ paired samples, the correct match for query $i$ is defined as the paired sample at index $i$. Recall@$k$ (R@1, R@5, R@10) is computed as the fraction of queries for which the correct match appears within the top-$k$ ranked candidates. Retrieval is evaluated in both directions for each modality pair (e.g., TS→IMG and IMG→TS), with rankings computed independently per direction. No thresholds or learned classifiers are used; the metrics depend solely on the rank order induced by cosine similarity, which is invariant to any positive temperature scaling.

## G. Datasets

We evaluate alignment using four datasets which we use to isolate the roles of semantic explicitness, visual mediation, and temporal complexity. All datasets are processed into aligned triplets of *time series*, *visual plots*, and *text*, with

consistent train–test splits and identical preprocessing across modalities. Because naturally occurring trimodal datasets are extremely scarce for the three modalities that we study, we complement available data with carefully generated synthetic and human-authored modalities to complete missing components wherever required.

## G.1. CaTS

CaTS is a multimodal dataset and benchmark which we use to study how numeric time series align with vision and language under controlled direct semantic conditions. Each sample consists of an univariate time series, a rendered plot of that series, and a natural language caption that describes observable temporal structure. The dataset spans 11 domains, including agriculture, air quality, border crossings, COVID statistics, crime, demography, diet, online retail, road injuries, $CO_2$, and Walmart sales. Figure 18 shows a representative CaTS triplet. We use the original CaTS training split (16k) for all training experiments. Evaluation is performed on held-out test sets for each caption variant, with a fixed validation set of 1k samples (which we generate following the same data generation framework used for CaTS). All variants share identical time series and plots, differing only in textual descriptions.

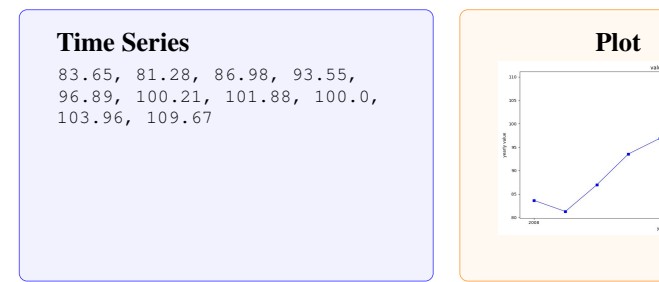

*Figure 18.* Example CaTS triplet consisting of a numeric time series, its visual plot, and a reference caption.

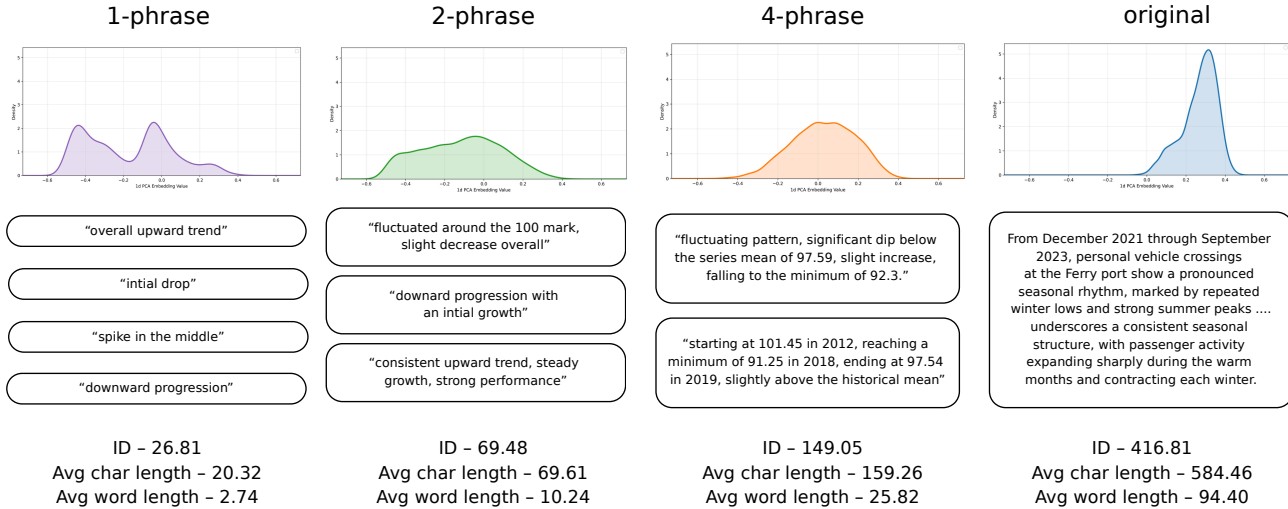

*Figure 19.* KDEs over projected embedding values and caption examples across CaTS variants with increasing ID.

To study the effect of semantic expressiveness, we generate multiple caption ID variants derived from the same underlying time series, plot, and metadata. Figure 19 illustrates distributional shifts in caption embeddings, visualized as kernel density estimates (KDEs) over projected embedding values. Starting from the original captions, we create progressively condensed versions containing approximately one phrase, two phrases, or four phrases. The condensed caption variants are generated only for evaluation on test samples. All variants are generated using a fixed protocol that preserves numeric fidelity while controlling verbosity. Caption extraction/summarization are performed using two LLMs, GPT-4o-mini and LLaMA-3.2-90B, with identical instructions applied across all samples. For these condensed variants, we instruct the language models to compress the original caption to a target number of semantic phrases, prioritizing the most salient temporal patterns, trends, and numeric attributes while discarding secondary details. This phrase-level constraint enables controlled reduction of

semantic explicitness without altering the underlying grounding or introducing new information. All synthetically generated caption variants are validated following the same verification protocol used in the original CaTS construction.

Next to study whether substantially increasing semantic richness during training can further improve multimodal alignment, we construct a High-ID variant of CaTS for both training and test splits. This dataset preserves the original time series and visual plots and we only modify the textual modality. High-ID captions are designed to be significantly more detailed than the original captions while maintaining strict numeric fidelity. Captions are generated independently for each sample using large language models (GPT-4o-mini and LLaMA-3.2-90B). Each model receives only the sample-specific metadata, and raw time series values. All captions are generated with low temperature to ensure determinism and consistency and all captions are generated using the following fixed system prompt, which explicitly enforces exhaustive numerical reporting, pattern and phase-wise analysis, and strict grounding in the provided data. The resulting High-ID captions are substantially longer and more semantically dense than the original captions, exceeding them by more than a factor of two in measured ID.

```
Write an EXTREMELY detailed and comprehensive analysis of this time series data.
Time Series: <ts>
Metadata: <metadata>
Your caption should include MOST of the following in 5-7 sentences:

1. CONTEXT: Full description of what this data represents, including location, measurement type, and exact time
     period covered
2. STATISTICS: Exact numerical values for mean, minimum, maximum, and standard deviation, with explicit comparisons
     to historical norms
3. OVERALL TREND: Primary direction and magnitude of change across the entire series
4. PHASE ANALYSIS: Break the series into distinct phases/segments, describe each with specific values and date
     ranges
5. PATTERNS: Any cyclical, seasonal, or periodic behavior with specific period lengths
6. ANOMALIES: Any unusual spikes, drops, or deviations with exact positions in the timeline and their values
7. COMPARISONS: Explicit numerical comparisons of this series to historical/global averages
8. TEMPORAL DETAILS: Specific dates, years, months, or time ranges for all key events and transitions

CRITICAL RULES:
- Be exhaustive and thorough
- Include specific numbers for EVERY claim
- Do not use speculative language - only describe what is evident in the data
- Do not summarize briefly - provide full detail
- Minimum 150 words
- REMEMBER NOT ALL INFORMATION IS NEEDED TO BE PRESENT IF IT IS NOT EVIDENT IN THE DATA.
```

## G.2. TRUCE

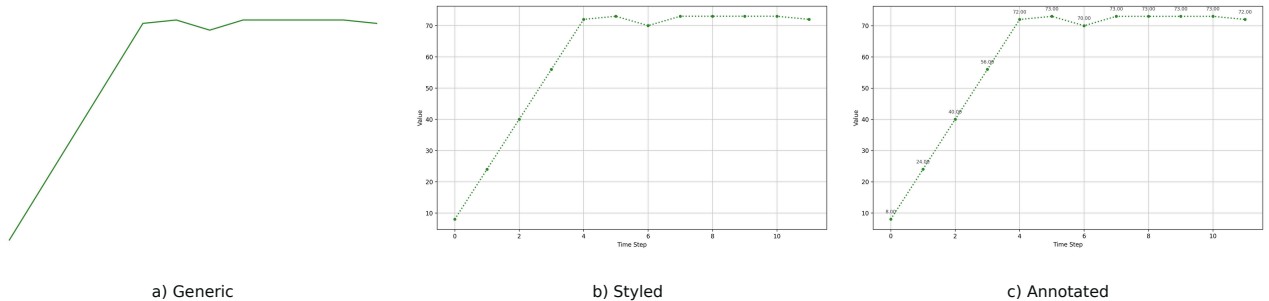

a) Generic          b) Styled          c) Annotated

*Figure 20.* Representative TRUCE visual variants: generic, styled, and annotated plots.

TRUCE is a controlled dataset which we use to isolate the effect of visual mediation on time series alignment. It consists of short, fixed-length univariate time series paired with concise direct textual descriptions. All time series have a length of 12 timesteps which enables exact visual annotation of every point. The training split consists 1,968 samples and the test split consists 492 samples. Representative TRUCE captions include "Increases linearly throughout the entirety of the graph," "Steady increase throughout," and "Troughs near the beginning."

Captions are short (approximately 85 characters on average) and directly describe local temporal structure, patterns, and trends. For each time series, we generate multiple visual variations as illustrated in Figure 20: *(a)* a generic plot showing only the raw time series curve, with no axes, annotations, or semantic markers, *(b)* a styled plot with randomized visual properties (line styles, colors, markers), and *(c)* an annotated plot in which every time step is labeled with its numeric value

and has styling.

## G.3. MIMIC-IV ECG

The MIMIC-IV ECG dataset provides real-world clinical time series paired with unstructured diagnostic reports. Each sample consists of a full-length ECG waveform and an associated clinical note that does not explicitly describe waveform structure. We use 21,000 samples with non-empty reports and valid waveform files. Time series is of 5,000 steps per record. Waveforms are rendered as line plots with axes and units. Figure 21 illustrates a triplet from MIMIC. We randomly shuffle the dataset and split it into 16,000 training samples and 4,000 test samples, with a fixed validation set.

All splits are disjoint at the subject–study level. Clinical reports are concatenated across available fields and used verbatim, without summarization or restructuring. As a result, the text modality provides indirect semantic supervision, focusing on diagnoses and observations rather than explicit temporal descriptions. MIMIC also helps in analysis of alignment when time series are substantially longer than those in CaTS or TRUCE.

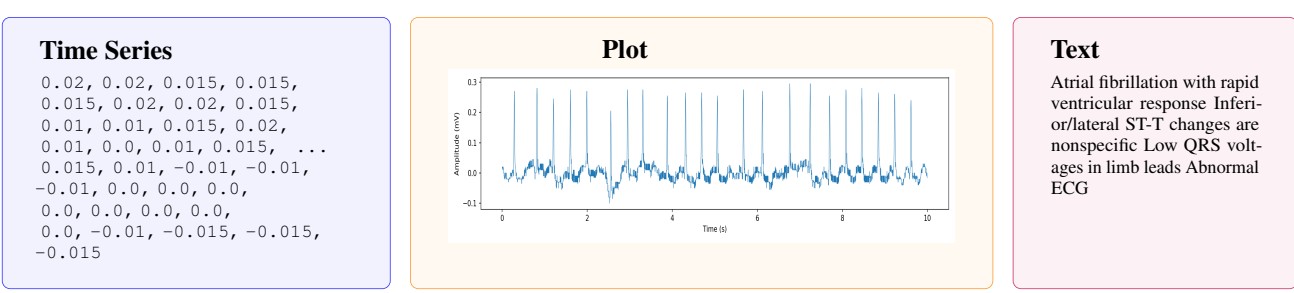

*Figure 21.* Example MIMIC triplet consisting of an ECG time series, its waveform visualization, and a report.

## G.4. PTB-XL

PTB-XL is a another ECG dataset which we use with standardized waveform recordings and diagnostic reports written primarily in German. Each recording is a fixed-length 10-second ECG sampled at 500 Hz, giving time series of length 5,000. Figure 22 illustrates a triplet from PTB-XL. We select samples with non-empty reports and valid waveform files, resulting in 21,000 aligned triplets. As with MIMIC, we render the full waveform with axes and units. Reports are used verbatim and remain in German, introducing a multilingual setting. The text modality again provides indirect semantic supervision, referring to clinical conditions rather than explicitly describing waveform structure.

PTB-XL allows us to disentangle the effects of indirect grounding from language mismatch, which complements MIMIC by holding the signal structure constant while varying linguistic properties. For PTB-XL, we evaluate a subset of configurations that use Qwen embedding models for the text modality. Qwen embeddings provide multilingual support, which is appropriate for the multilingual clinical text found in PTB-XL.

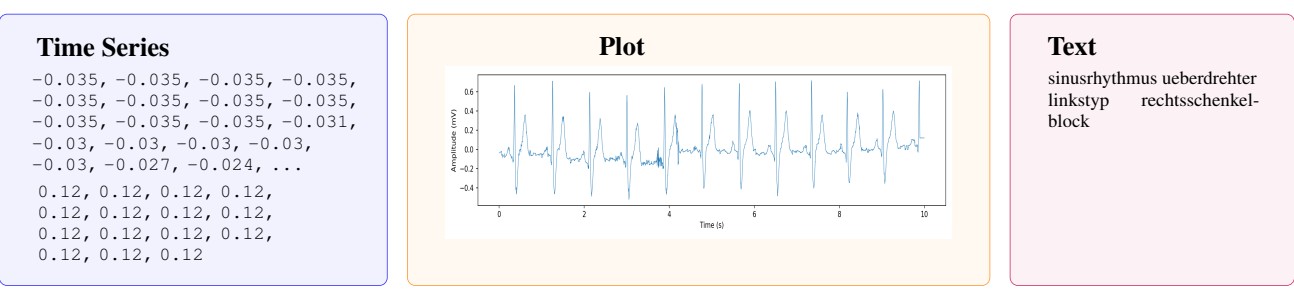

*Figure 22.* Example PTB-XL triplet consisting of an ECG time series, its waveform visualization, and a report.

# H. Information Density

We define *information density* as the total surprisal of a caption under a fixed pretrained language model. Given a caption $x = (w_1, \ldots, w_T)$, we compute the per-token surprisal

$$\bar{\ell}(x) = \frac{1}{T} \sum_{t=1}^{T} -\log p(w_t \mid w_{<t}), \tag{22}$$

and define

$$\text{ID}(x) = T \cdot \bar{\ell}(x), \tag{23}$$

which corresponds to the total negative log-likelihood. Surprisal is computed using a pretrained GPT-2 language model with frozen parameters.

There is no model-agnostic metric that directly measures the amount of semantic content expressed in unconstrained natural language. Motivated by information–uniformity perspectives on language modeling (Jaeger & Levy, 2006; Meister et al., 2021), we use language-model surprisal as a principled proxy for *expressed information* under a fixed linguistic prior. Intuitively, natural language text that convey more specific, grounded content require more bits to encode and thus exhibit higher surprisal, whereas generic or templated descriptions are more predictable. While surprisal is not a perfect semantic measure, it provides a controlled and scalable operationalization for comparing text sets without introducing task- or model-specific heuristics.

Our use of ID is also consistent with how PRH operationalizes textual richness. They show that vision–language alignment improves by increasing caption density via longer and more descriptive captions, on the Densely-Captioned-Images (DCI) dataset (Urbanek et al., 2024). In this, caption length effectively serves as a proxy for textual density. We follow the same intuition, but make the notion of density more explicit and interpretable by jointly accounting for caption length and per-token surprisal under a fixed language model. This allows us to distinguish between captions that are merely longer and those that express more information, while remaining aligned with the PRH perspective.

**Length and Per-Token Controls.** Because total surprisal scales with sequence length by construction, we explicitly report caption length, tokens, words alongside ID. Table 6 summarizes these statistics for the caption variants used in our experiments.

*Table 6.* ID with explicit length and words controls (averaged over test captions).

| Caption Variant | Tokens ($T$) | Words | ID |
| --- | --- | --- | --- |
| 1-phrase | 3.14 | 2.74 | 26.81 |
| 2-phrase | 14.01 | 10.24 | 69.48 |
| 4-phrase | 37.22 | 25.82 | 149.05 |
| Dense | 128.73 | 94.40 | 416.81 |
| High-ID | 312.28 | 231.75 | 867.40 |

*Table 7.* Stability of ID ordering across GPT variant language models.

| Caption Variant | GPT-2 | GPT-2 Medium | GPT-2 Large |
| --- | --- | --- | --- |
| 1-phrase | 1 | 1 | 1 |
| 2-phrase | 2 | 2 | 2 |
| 4-phrase | 3 | 3 | 3 |
| Dense | 4 | 4 | 4 |
| High-ID | 5 | 5 | 5 |

**Robustness to Language Model Choice.** To assess sensitivity to the surprisal model, we recompute ID using GPT-2, GPT-2 Medium, and GPT-2 Large, keeping all other settings fixed. Across all three models, caption variants exhibit an identical ordering in information density (*1-phrase < 2-phrase < 4-phrase < dense < high-ID*), as shown in Table 7. At the caption level, ID values show high rank agreement across models, and all downstream alignment analyses exhibit the same qualitative trends. This indicates that ID provides a stable and model-robust axis for comparing caption variants.

# I. Line Plots as Visual Projections of TS and TXT

A natural concern is whether the observed asymmetry between TS–IMG and TS–TXT alignment is an artifact of using visual plots that are direct renderings of the underlying time series. Because line plots are deterministic transformations of the signal, TS–IMG alignment may appear easier than TS–TXT, potentially amplifying the observed gap. We therefore discuss here whether the observed behavior reflects trivial pixel-level correspondence or deeper structural compatibility

across modalities. Under the PRH, different modalities are viewed as projections of a shared latent reality. For time series, visual plots provide a structured projection that externalizes latent temporal properties, such as trends, extrema, and phase transitions, into explicit geometric form. This transformation does not introduce new information, but it does reorganize existing signal structure into a representation that is more directly comparable to spatial and symbolic modalities.

To test whether TS–IMG alignment is driven by superficial visual matching, we also evaluate robustness under nontrivial visual perturbations as shown in Figure 20. Using TRUCE, we compare generic, stylistically varied, and annotated plots while keeping the underlying time series and text fixed. If alignment were purely tautological, changes in visual realization would either have little effect across variants. Instead, we observe that TS–IMG alignment does not collapse under style variation and is consistently higher for styled and annotated plots than for generic renderings which shows that alignment tracks structured signal cues rather than brittle pixel correspondence.

In contrast, TS–TXT alignment remains consistently weaker even when captions directly describe the same temporal structure. Styled and annotated plots further improve TS–IMG alignment by making implicit signal attributes explicit. However, the central observation is not the absolute gain from annotation, but the robustness of TS–IMG alignment across visually distinct yet semantically equivalent projections. This shows that visual plots function as structured representations that externalize temporal structure in a form that is naturally compatible with contrastive alignment. This asymmetry does not appear to be tied to a particular visual encoding and it persists across diverse *direct* visual realizations of the same underlying signal.

We further identify which visual annotation components contribute most strongly to the TS–IMG alignment. Generic plots, consisting of plain line curves, exhibit the lowest alignment scores. Styled plots, which introduce variations in color, line style, and markers, show substantial improvements in most configurations. Annotated plots achieve the highest alignment by incorporating explicit numeric labels at each timestep along with labeled axes, which indicates that numeric grounding along with stylistic markers is the dominant factor driving improved alignment.

To assess generalizability beyond line plots, we evaluate recurrence plots as an alternative visual encoding. Recurrence plots are visually very different from line plots yet retain explicit temporal structure in geometric form. As shown in Table 8, TS–IMG alignment remains strong and consistently higher than TS–TXT across both encodings, suggesting the effect reflects externalization of temporal semantics rather than dependence on a specific visual convention.

*Table 8.* Procrustes disparity ($\downarrow$) for alternative visual encodings compared to TS–TXT alignment.

| Configuration | TS–Line | TS–Recurrence | TS–TXT |
|---|---|---|---|
| DINOv2-B + T5-Small + Chronos-B | 0.49 | 0.57 | 0.68 |
| DINOv2-B + E5-Base + TimesFM-B | 0.65 | 0.62 | 0.76 |

As a further control, we evaluate temporally scrambled plots, generated by randomly permuting the time axis of each series while keeping all visual properties identical, including axes, colors, line style, and rendering. This intervention preserves the visual format while destroying temporal semantics. If TS–IMG alignment were driven by trivial pixel-level correspondence between a signal and its plot, scrambling would have little effect. Instead, we observe a sharp degradation across four lower-bound DINOv2 configurations: cosine similarity decreases by 79.5% and Procrustes distance increases by 210.7%. This further indicates that alignment is sensitive to temporal semantics and is not an artifact of the deterministic relationship between a time series and its visual rendering.

All these controls argue against structural similarity as the primary driver of TS–IMG alignment. First, recurrence plots, which are visually very different from line plots, maintain strong TS–IMG alignment. Second, temporally scrambled plots, which are visually identical to standard plots but have destroyed temporal semantics, collapse alignment substantially. Third, progressively richer visual annotations improve TS–IMG alignment, showing that alignment tracks semantic content rather than mere structural correspondence. Fourth, more explicit textual descriptions improve TS–TXT alignment but TS–IMG remains consistently higher, indicating the gap arises from how modalities encode temporal semantics rather than from trivial structural similarity between TS and their plots.

An interesting open question is how these observations extend to more implicit visual encodings of temporal phenomena. For example, consider a physics simulation in which a ball rolls down a ramp: the ball's trajectory can be represented as a time series, individual frames of the simulation as the visual modality, and text can describe both the evolving dynamics and summary statistics of the motion. Such settings would allow studying alignment when visual structure emerges from

physical dynamics rather than explicit plotting conventions. To the best of our knowledge, however, no existing datasets provide well-aligned time series–image–text triplets of this form, and we leave this direction to future work.

## J. Representative Results of Trimodal Configurations

**Effect of Scaling All Three Encoders.**  Figure 23 shows that increasing the capacity of all three encoders leads to consistent improvements in cross-modal alignment, but with marked asymmetry across modality pairs. Alignment involving time series improves most strongly for TS–IMG across all metrics, while gains for TS–TXT are smaller and less consistent. Notably, geometric metrics and cosine margins exhibit similar trends which indicates that scaling improves both global geometry and local neighborhood structure. This shows that while model scale improves alignment, the extent of improvement depends on the semantic compatibility of the modality pair, with TS-TXT remaining the most challenging even at larger scales.

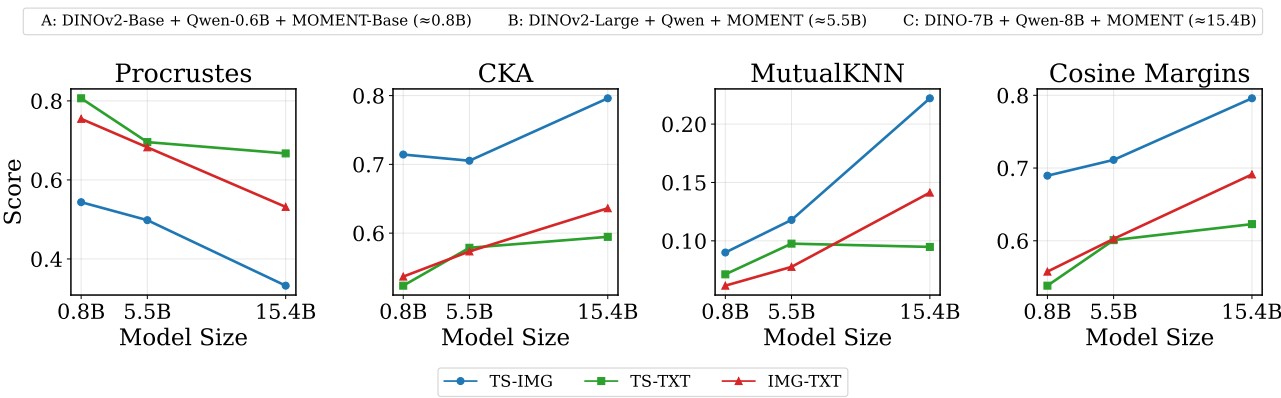

*Figure 23.* Effect of joint encoder scaling on trimodal alignment. Alignment improves with model size across modality pairs, but gains are uneven. When the time series encoder is already saturated (from B to C), additional scaling yields substantial improvements for TS–IMG alignment, while TS–TXT exhibits smaller and less consistent changes across metrics. This indicates that TS–TXT alignment remains more challenging than TS–IMG: scaling the vision encoder produces clear gains for TS–IMG, whereas corresponding improvements to the text encoder lead to only modest or saturating gains for TS–TXT.

**Effect of Time Series Encoder Capacity.**  To isolate the role of time series encoder strength, we scale the time series encoder while holding vision and text encoders fixed. Figure 24 compares TimesFM-200M and TimesFM-500M paired with EVA-CLIP (8B) and Mistral (12B). Increasing time series encoder capacity leads to consistent improvements in TS–IMG and TS–TXT alignment across cosine similarity, retrieval metrics, and Procrustes geometry. IMG–TXT alignment shows only minimal improvement.

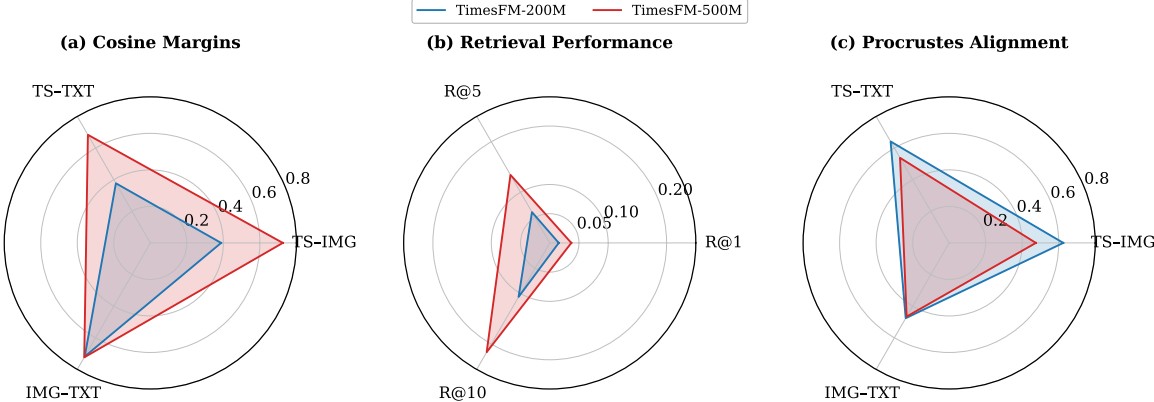

*Figure 24.* Comparison of multimodal alignment using TimesFM-200M and TimesFM-500M time series encoders, with EVA-CLIP (8B) and Mistral (12B) held fixed. **(a)** Cosine margins for TS–IMG, TS–TXT, and IMG–TXT. **(b)** Retrieval performance (R@1, R@5, R@10). **(c)** Procrustes alignment (lower is better).

## K. Human Linguistic Shifts

We compare alignment obtained using synthetic captions versus human-written captions from the CaTS human test set, evaluated with CaTS-trained models, as shown in Figure 25 for a small set of representative domain samples (two domains). Both settings operate in the direct text representation regime. Across configurations and modality pairs, we observe only modest shifts in alignment when replacing synthetic captions with human-authored descriptions. In some configurations, alignment slightly improves under human captions, while in others it exhibits a small degradation; however, these effects are consistently limited in magnitude across both cosine similarity and Procrustes-based geometry. Human captions exhibit higher ID on the 2 domain test set (409.15 vs. 381.95), inducing a mild out-of-distribution linguistic shift relative to the training distribution. Despite this shift, alignment remains largely stable, indicating that the learned trimodal embedding spaces generalize well to natural human linguistic variability. The modest difference between LLM and human captions suggests that the shift is insufficient to induce collapse or improvement which also further support the generalization of models trained on CaTS to human-authored descriptions.

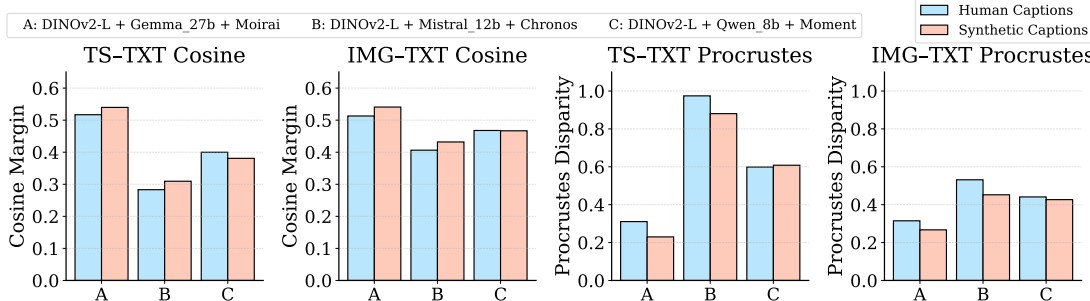

*Figure 25.* Alignment comparison between LLM-generated and human-written captions across modality pairs.

## L. Discussion on Semantic Explicitness for Alignment Asymmetry

Our findings show that trimodal alignment is not uniform: TS–IMG alignment consistently outperforms TS–TXT, and images can act as effective intermediaries between time series and text. We use the term *semantic explicitness* in an operational, descriptive sense to interpret this asymmetry, referring to the degree to which a modality's encoding makes underlying semantic structure directly observable rather than requiring abstraction or inference.

Time series encode temporal properties implicitly. Characteristics like trends, periodicity, or anomalies are not directly observable but they must be inferred through computation over raw numerical values. Recognizing an "increasing trend" in a sequence $[v_1, v_2, \ldots, v_T]$ requires computing differences, aggregating, and applying some decision criterion. The time series encoder must learn to perform this inference from data. Visual plots make this structure explicit. A trend manifests as a visible slope, a peak becomes a geometric feature, and periodicity appears as repeating spatial patterns. The transformation from time series to plot largely preserves structure while externalizing latent patterns into directly observable form. Vision encoders, with their inductive biases for detecting edges, slopes, and shapes, can use this explicit structure directly.

Text occupies a different position. Captions explicitly *name* semantic concepts such as the phrase "increasing trend" directly references the property but this naming is abstract. The symbol "trend" can refer to infinitely many numerical realizations; it provides no grounding in specific values or shapes. A caption describes *what* is present without encoding *how* it appears numerically or visually. This gap may explain why TS–TXT alignment is weaker than TS–IMG in our experiments: matching implicit numerical patterns to abstract language is harder than matching them to structured visual representations. We believe this interpretation is consistent with the alignment patterns observed across our experiments.

**TS–IMG outperforms TS–TXT.** Across models and metrics, time series align more strongly with visual plots than with text, consistently exhibiting closer proximity to images than to language in the shared embedding space. Plots externalize temporal structure into explicit geometric form, whereas text abstracts over specific realizations, suggesting that *how* information is encoded matters as much as *what* is encoded.

**Images function as semantic intermediaries.** Introducing the image modality consistently improves TS–TXT alignment compared to bimodal training. This is consistent with images serving as explicit intermediates: time series can align with

plots (implicit → explicit), and plots can align with text (explicit → abstract), providing a pathway that circumvents the direct implicit-to-abstract mapping. The trimodal objective allows representations to leverage this intermediate grounding. Notably, this effect is also particularly strong for vision encoders pretrained with a VL objective (SigLIP), even when only their vision component is used. This suggests that VL pretraining shapes image representations to be more semantically compatible with text by inducing shared geometric structure in the embedding space (Radford et al., 2021; Levi & Gilboa, 2025), further strengthening the role of images as effective semantic bridges in trimodal alignment.

To quantify this geometric intermediacy directly, we test whether image embeddings lie geometrically between time series and text embeddings in the shared space. For each TS–IMG–TXT triplet, we check whether the image embedding is closer to both endpoints than they are to each other in configuration 12 in Table 4. We find that $73.4\%$ of triplets satisfy this condition, confirming that images occupy a geometrically intermediate position. We also measure this using a triangle-chain error, defined as the mean relative residual when the image is used to chain TS and TXT. The mean error drops from $0.90$ for DINOv2 configurations to $0.34$ for SigLIP2 configurations, indicating that VL pretraining shapes image representations to be more semantically compatible with both modalities simultaneously, further strengthening their role as semantic bridges.

**Information density shows diminishing returns.** Increasing caption richness improves alignment at low-to-moderate levels, but further increases yield marginal gains. This suggests that while richer text increases specificity which effectively raising its semantic explicitness, the abstraction inherent to symbolic representation limits further improvement. Text can become more detailed, but it remains symbolic; it cannot become a continuous representation of the data.

**Indirect text degrades alignment.** Clinical reports in MIMIC describe diagnostic conclusions ("atrial fibrillation") rather than waveform structure ("sharp peak followed by gradual decay"). This further abstraction widens the explicitness gap, and alignment suffers accordingly.

**Semantic explicitness as a controlled ablation.** To directly test whether the abstraction gap of text contributes to weaker TS–TXT alignment, we perform a targeted caption ablation in which we replace the original CaTS captions sets with structured, explicitly grounded descriptions that bind language directly to numeric values, temporal ranges, and phase structure (e.g., explicit start/end values, monotonic changes, and segmented trends). Importantly, these structured captions are designed to be more explicit while maintaining ID comparable to the original CaTS captions, and remain far below the high-ID regime that exhibited saturation. This intervention is designed as a controlled diagnostic that isolates one concrete dimension of textual grounding while holding overall ID approximately constant.

As shown in Table 9, more explicitly grounded captions provide improvements in TS–TXT alignment across all metrics, with parallel gains for IMG–TXT alignment. These results further support the interpretation that alignment asymmetry is driven in part by differences in how explicitly semantic structure is encoded across modalities. Prompt is provided below.

```
Write a CLEAR, EXPLICIT, and STRUCTURALLY GROUNDED description of this time series.

GOAL:
Reduce abstraction while keeping natural, grammatical English.
Bind language directly to numeric and temporal structure.

FORMAT REQUIREMENTS:
- Natural English, full sentences
- 4-5 sentences
- Maximum 200 words total
- Concise but explicit (do NOT be verbose)

INCLUDE:
1. CONTEXT: What the data represents, including location, metric, and exact time span.
2. NUMERIC ANCHORS: Explicit start, end, minimum, and maximum values, each with time positions.
3. TEMPORAL STRUCTURE: Describe changes using explicit numeric phrasing, e.g.:
    - "increases monotonically from X to Y"
    - "drops from X to Y between year A and B"
4. PHASES: If the series naturally decomposes, describe up to 2-3 phases. Each phase must include a numeric value
     range and a time range.
5. PATTERNS: Use explicit terms ("monotonic increase", "single spike at midpoint") instead of abstract descriptors.

CRITICAL RULES:
- Every trend or pattern claim MUST reference specific numbers
- Do NOT use qualitative adjectives (e.g., steady, sharp, notable, gradual) unless immediately quantified
- Do NOT speculate, explain causes, or add interpretations
- Do NOT add information not present in the data
```

*Table 9.* Explicit semantic grounding improves text alignment at comparable ID. Comparison of original CaTS captions (train ID = 417.2) and structured captions (train ID = 405.8) for a representative trimodal configuration (DINOv2 + Qwen + MOMENT). All encoders and training settings are identical; only the captions differ.

| Caption Type | Pair | Cosine Margin ↑ | Procrustes ↓ | CKA ↑ |
|---|---|---|---|---|
| CaTS (original) | TS–TXT | 0.538 | 0.807 | 0.524 |
| Structured | TS–TXT | **0.574** | **0.754** | **0.568** |
| CaTS (original) | IMG–TXT | 0.558 | 0.755 | 0.537 |
| Structured | IMG–TXT | **0.588** | **0.714** | **0.583** |

**Code as an Alternative Textual Modality.** A natural question is whether using code as the textual modality, rather than natural language (NL), could reduce the TS–TXT alignment gap. Code explicitly specifies the data transformations and plotting logic that generate a time series visualization, occupying a middle ground between the implicit structure of time series and the symbolic abstraction of natural language. We evaluate two configurations: Config A (DINOv2-B + Qwen-0.6B + MOMENT-B) and Config B (ViT-B + T5-Small + MOMENT-B), comparing natural language captions against Python code descriptions of the same time series.

As shown in Table 10, code representations improve TS–TXT alignment for Config A (Qwen), suggesting that code may place as an intermediate point on the explicitness axis: TS–IMG > TS–TXT (code) > TS–TXT (NL). For Config B (T5), code provides minimal benefit, likely because T5 was pretrained primarily on natural language, limiting its ability to parse structured code semantics. Importantly, the core asymmetry persists across both configurations, consistent with our hypothesis that more explicit representations align better while the fundamental TS–IMG advantage remains.

*Table 10.* Effect of code vs. natural language as the textual modality. Config A: DINOv2-B + Qwen-0.6B + MOMENT-B. Config B: ViT-B + T5-Small + MOMENT-B. Procrustes Disparity (PD) and Cosine Margin (Cos) are used to measure alignment.

| Config | Text | PD TS–IMG ↓ | PD TS–TXT ↓ | Cos TS–IMG ↑ | Cos TS–TXT ↑ | CKA TS–IMG ↑ | CKA TS–TXT ↑ |
|---|---|---|---|---|---|---|---|
| A | NL | 0.5437 | 0.8066 | 0.6895 | 0.5383 | 0.7143 | 0.5236 |
| A | Code | 0.5412 | 0.6234 | 0.6901 | 0.6234 | 0.7168 | 0.6412 |
| B | NL | 0.5516 | 0.7604 | 0.6843 | 0.5692 | 0.7326 | 0.5554 |
| B | Code | 0.5501 | 0.7689 | 0.6851 | 0.5698 | 0.7334 | 0.5563 |

These observations suggest a refinement to how multimodal convergence with time series should be understood. The PRH predicts that representations of the same underlying structure will converge. Our findings indicate this convergence may be conditional: in our experiments, modalities that encode semantics and structure with similar explicitness align more readily, while those with larger explicitness gaps show weaker alignment. Rather than converging uniformly toward a single shared representation, modalities appear to align to different degrees depending on their representational format.

# M. Limitations and Future Directions

In this section, we elaborate on the limitations and future directions of our work. Several questions remain open for future research. Our main goal was to provide a first systematic study of trimodal alignment in contrastive spaces involving time series, vision and language, rather than a definitive characterization of all possible settings.

Our analysis focuses on univariate time series, and it remains an open question whether the alignment behaviors observed here extend to multivariate signals. CaTS captions are synthetic, which may not fully capture the diversity of human writing styles. We partially address this by evaluating on CaTS human-written, directly descriptive captions but is limited in scale. While we also test on real clinical text from MIMIC and PTB-XL, these datasets are domain-specific and rely on indirect language, which constrains their generalization for studying direct textual grounding. Also to ensure controlled and fair comparison across a large number of encoder combinations, we adopt a fixed contrastive training protocol with frozen encoders and train projection heads. This design isolates intrinsic representational compatibility across modalities, but precludes studying how end-to-end fine-tuning or modality-specific adaptation might further shape alignment. Exploring such training regimes would be a natural extension, but doing so exhaustively across all configurations considered in this work would be computationally prohibitive. As a further direction, it would also be valuable to evaluate the proposed analysis on sensor-based time series data, such as IMU-based datasets (e.g., Ego4D (Grauman et al., 2022)), which would

enable comparison with existing sensor-based image–text–time series alignment methods such as ImageBind (Girdhar et al., 2023) and IMU2CLIP (Moon et al., 2023). This evaluation could help determine whether the observed alignment behavior generalizes beyond the current setup constructed.

An important direction for future work is to investigate how the alignment phenomena that we identify correlate with downstream task performance in multimodal time series applications. However, large-scale datasets that simultaneously include aligned time series, visual representations, and natural language supervision remain scarce, which limits systematic evaluation of task-level transfer. We hope that our findings highlight the need for richer multimodal time series datasets and stimulate further development, helping in more studies of how representational alignment in this trimodal setting translates into practical performance gains.

Developing formal theoretical frameworks that explain the observed findings remains another important open direction. In our work we prioritize rigorous experimentation to isolate key factors governing alignment behavior, including scale, geometry, information density, and modality-specific characteristics. While extensive prior work has explored VL alignment, our goal is to establish a comparable empirical foundation for multimodal time series data and encourage further study in this emerging area. Providing a unified theory of multimodal representations remains challenging in general, particularly for numerically implicit modalities such as time series.

We provide strong conceptual grounding for the observed phenomena through multiple hypotheses, analyses, and controlled experiments. In particular, throughout the main paper and Appendices L, I, C we discuss interpretations of the observed behaviors and connect them to intrinsic properties of the underlying data and modalities. Across our study, we identify several previously unreported phenomena for our studied settings, including alignment asymmetry, scaling behavior, information density saturation, image bridging effects, and global–local dissociation in contrastive representation spaces. We believe that theory and experimentation in multimodal learning should progress iteratively, and we view the empirical findings identified here as a necessary starting point toward developing more formal theoretical frameworks for multimodal representation alignment.

## N. Full Results

### N.1. Results on varying ID text on CaTS models

Tables 11 and 12 provide the full numerical results underlying Figure 7, which summarizes the effect of textual ID on TS–TXT and TXT–IMG alignment across models and caption distributions.

*Table 11.* TS–TXT alignment metrics across four caption distributions (1-phrase, 2-phrase, 4-phrase, and dense).

| Model | Cosine Margin | | | | Procrustes | | | | Mutual KNN | | | |
|---|---|---|---|---|---|---|---|---|---|---|---|---|
| | 1-phrase | 2-phrase | 4-phrase | dense | 1-phrase | 2-phrase | 4-phrase | dense | 1-phrase | 2-phrase | 4-phrase | dense |
| eva_18b + mistral_12b + moment | 0.0630 | 0.1371 | 0.2940 | 0.6631 | 3.0968 | 1.9891 | 1.3428 | 0.5785 | 0.0165 | 0.0321 | 0.0436 | 0.1053 |
| eva_8b + mistral_12b + timesfm | 0.0747 | 0.1590 | 0.3116 | 0.6795 | 2.6904 | 1.8769 | 1.3067 | 0.5401 | 0.0189 | 0.0301 | 0.0481 | 0.1164 |
| dino_7b + gemma_27b + moirai | 0.0049 | 0.1014 | 0.3423 | 0.7432 | 3.3569 | 2.0266 | 1.2302 | 0.4370 | 0.0095 | 0.0188 | 0.0499 | 0.1451 |
| eva_18b + gemma_27b + chronos | 0.0467 | 0.1386 | 0.2684 | 0.6866 | 2.8463 | 1.8538 | 1.3591 | 0.5180 | 0.0125 | 0.0241 | 0.0419 | 0.1228 |
| siglip + t5 + timesfm | 0.1614 | 0.2753 | 0.4229 | 0.6583 | 1.6072 | 1.4018 | 1.0305 | 0.5827 | 0.0168 | 0.0300 | 0.0607 | 0.1025 |
| dinov2_base + t5_small + chronos | 0.1385 | 0.2183 | 0.3601 | 0.5966 | 1.6938 | 1.5143 | 1.1394 | 0.6825 | 0.0167 | 0.0229 | 0.0405 | 0.0787 |
| siglip + e5 + moment | 0.1920 | 0.2612 | 0.3913 | 0.6362 | 1.6170 | 1.4234 | 1.0947 | 0.6309 | 0.0188 | 0.0281 | 0.0513 | 0.1053 |
| vit_base + qwen_06b + moirai | 0.0400 | 0.1311 | 0.2883 | 0.5505 | 3.1128 | 2.0810 | 1.3682 | 0.8092 | 0.0093 | 0.0192 | 0.0263 | 0.0652 |
| vit_base + t5_small + moment | 0.1637 | 0.2275 | 0.3592 | 0.5692 | 1.6617 | 1.5262 | 1.1801 | 0.7604 | 0.0149 | 0.0289 | 0.0447 | 0.0664 |

*Table 12.* TXT–IMG alignment metrics across four caption distributions (1-phrase, 2-phrase, 4-phrase, and dense).

| Model | Cosine Margin | | | | Procrustes | | | | Mutual KNN | | | |
|---|---|---|---|---|---|---|---|---|---|---|---|---|
| | 1-phrase | 2-phrase | 4-phrase | dense | 1-phrase | 2-phrase | 4-phrase | dense | 1-phrase | 2-phrase | 4-phrase | dense |
| eva_18b + mistral_12b + moment | 0.0581 | 0.1409 | 0.3132 | 0.7275 | 3.0861 | 1.9440 | 1.2660 | 0.4613 | 0.0177 | 0.0269 | 0.0495 | 0.1180 |
| eva_8b + mistral_12b + timesfm | 0.0670 | 0.1557 | 0.3209 | 0.7330 | 2.7057 | 1.8688 | 1.2621 | 0.4517 | 0.0181 | 0.0323 | 0.0517 | 0.1165 |
| dino_7b + gemma_27b + moirai | 0.0125 | 0.1085 | 0.3235 | 0.7581 | 3.1842 | 1.8941 | 1.1987 | 0.3874 | 0.0169 | 0.0228 | 0.0561 | 0.1804 |
| eva_18b + gemma_27b + chronos | 0.0564 | 0.1605 | 0.3159 | 0.7318 | 2.7981 | 1.7660 | 1.2222 | 0.4331 | 0.0159 | 0.0276 | 0.0576 | 0.1409 |
| siglip + t5 + timesfm | 0.1655 | 0.2888 | 0.4805 | 0.7627 | 1.5824 | 1.3488 | 0.8888 | 0.3822 | 0.0207 | 0.0409 | 0.0924 | 0.1780 |
| dinov2_base + t5_small + chronos | 0.1392 | 0.2238 | 0.3820 | 0.6237 | 1.6949 | 1.4807 | 1.0812 | 0.6451 | 0.0129 | 0.0215 | 0.0413 | 0.0643 |
| siglip + e5 + moment | 0.1943 | 0.2743 | 0.4285 | 0.7150 | 1.5917 | 1.3586 | 0.9767 | 0.4653 | 0.0240 | 0.0340 | 0.0761 | 0.1617 |
| vit_base + qwen_06b + moirai | 0.0606 | 0.1479 | 0.2905 | 0.5398 | 2.9071 | 1.9151 | 1.2641 | 0.7750 | 0.0151 | 0.0176 | 0.0351 | 0.0607 |
| vit_base + t5_small + moment | 0.1279 | 0.2080 | 0.3692 | 0.6156 | 1.7234 | 1.5262 | 1.1192 | 0.6702 | 0.0121 | 0.0235 | 0.0447 | 0.0711 |

## N.2. Results on CaTS test set

*Table 13.* Correlation between alignment metrics and cross-modal retrieval performance. Mutual $k$NN overlap shows the strongest and most consistent association with retrieval performance, particularly for TS–IMG pairs, with substantial correlations also observed for IMG–TXT and TS–TXT. Procrustes disparity exhibits strong negative correlations, indicating that improved global geometric alignment (lower disparity) corresponds to higher retrieval accuracy. Cosine similarity and CKA show weaker but still meaningful correlations, especially for harder TS–TXT pairs. Overall, higher alignment quality corresponds to improved cross-modal retrieval performance.

| Metric | Pair | Pearson $r$ | | | Spearman $\rho$ | | |
|---|---|---|---|---|---|---|---|
| | | R@1 | R@5 | R@10 | R@1 | R@5 | R@10 |
| Cosine Similarity | TS–IMG | 0.8333 | 0.8489 | 0.8544 | 0.8834 | 0.8814 | 0.8762 |
| | TS–TXT | 0.3166 | 0.3691 | 0.3974 | 0.3348 | 0.3600 | 0.3780 |
| | IMG–TXT | 0.6277 | 0.6718 | 0.6960 | 0.6443 | 0.6753 | 0.6826 |
| Procrustes Disparity | TS–IMG | -0.8360 | -0.8536 | -0.8604 | -0.9046 | -0.8972 | -0.8870 |
| | TS–TXT | -0.3951 | -0.4512 | -0.4821 | -0.4654 | -0.4865 | -0.5007 |
| | IMG–TXT | -0.6179 | -0.6612 | -0.6852 | -0.6432 | -0.6684 | -0.6734 |
| CKA | TS–IMG | 0.7217 | 0.7092 | 0.7014 | 0.7426 | 0.7347 | 0.7170 |
| | TS–TXT | 0.2585 | 0.3100 | 0.3373 | 0.2821 | 0.3100 | 0.3292 |
| | IMG–TXT | 0.5064 | 0.5433 | 0.5664 | 0.5410 | 0.5570 | 0.5595 |
| Mutual kNN | TS–IMG | 0.9282 | 0.9332 | 0.9330 | 0.9410 | 0.9436 | 0.9353 |
| | TS–TXT | 0.4618 | 0.5194 | 0.5525 | 0.5325 | 0.5539 | 0.5746 |
| | IMG–TXT | 0.8332 | 0.8495 | 0.8541 | 0.7933 | 0.8153 | 0.8225 |

Tables 14 and 15 report the complete results corresponding to Figure 5, providing per-configuration alignment metrics across all model combinations and modalities. While TS–IMG alignment dominates in the majority of configurations, we observe a small number of cases (3) where TS–TXT exceeds TS–IMG. Notably, all such exceptions involve the Moirai time series encoder, suggesting a systematic pattern rather than noise. In these configurations, Moirai-based representations exhibit stronger compatibility with text embeddings than with visual embeddings. Although the internal causes of this behavior are not directly observable, one possible interpretation is that Moirai encodings produce higher-level temporal abstractions that align more readily with symbolic language than with the fine-grained visual cues used for plots. Overall these cases highlight that alignment behavior is not identical across all time series encoders and can depend on architectural interactions in trimodal settings.

*Table 14.* Cross-modal evaluation reporting cosine similarity margins and retrieval performance across model configurations.

| Model Configuration | Cosine Similarity Margins | | | Cross-Modal Retrieval | | | |
|---|---|---|---|---|---|---|---|
| | TS–IMG | TS–TXT | IMG–TXT | R@1 | R@5 | R@10 | MRR |
| *Lower Bound Models* | | | | | | | |
| dinov2-b + t5_small + moirai-b | 0.5835 | 0.6478 | 0.5902 | 0.0081 | 0.0358 | 0.0689 | 0.0336 |
| vit-b + t5_small + moment-b | 0.6843 | 0.5692 | 0.6156 | 0.0197 | 0.0763 | 0.1279 | 0.0597 |
| dinov2-b + t5_small + chronos-b | 0.7082 | 0.5966 | 0.6237 | 0.0239 | 0.0846 | 0.1400 | 0.0660 |
| dinov2-b + e5-base + timesfm-b | 0.6273 | 0.5530 | 0.6275 | 0.0122 | 0.0501 | 0.0928 | 0.0433 |
| vit-b + e5-b + chronos-b | 0.7200 | 0.5729 | 0.6169 | 0.0218 | 0.0825 | 0.1365 | 0.0633 |
| siglip2-b + e5-b + moirai-b | 0.7050 | 0.5702 | 0.7328 | 0.0160 | 0.0640 | 0.1135 | 0.0535 |
| siglip2-b + e5-b + moment-b | 0.6734 | 0.5484 | 0.7205 | 0.0238 | 0.0868 | 0.1440 | 0.0677 |
| siglip2-b + t5_small + timesfm-b | 0.6292 | 0.5445 | 0.7236 | 0.0212 | 0.0816 | 0.1392 | 0.0648 |
| vit-b + qwen_06b + moirai-b | 0.6278 | 0.5505 | 0.5398 | 0.0097 | 0.0408 | 0.0725 | 0.0354 |
| dinov2-b + qwen_06b + moment-b | 0.6895 | 0.5383 | 0.5576 | 0.0161 | 0.0613 | 0.1075 | 0.0503 |
| vit-b + qwen_06b + timesfm-b | 0.6379 | 0.5162 | 0.5467 | 0.0137 | 0.0573 | 0.1021 | 0.0477 |
| siglip2-b + qwen_06b + chronos-b | 0.7147 | 0.5432 | 0.6758 | 0.0396 | 0.1264 | 0.1985 | 0.0940 |
| *Middle Bound Models* | | | | | | | |
| vit_h14 + t5 + moment-l | 0.7065 | 0.6521 | 0.6701 | 0.0283 | 0.1026 | 0.1692 | 0.0784 |
| siglip + t5 + timesfm-l | 0.7582 | 0.6583 | 0.7627 | 0.0588 | 0.1875 | 0.2851 | 0.1338 |
| dinov2 + t5 + moirai-l | 0.6044 | 0.7015 | 0.6249 | 0.0117 | 0.0500 | 0.0919 | 0.0437 |
| dinov2 + t5 + chronos-l | 0.7270 | 0.6397 | 0.6579 | 0.0311 | 0.1052 | 0.1720 | 0.0800 |
| vit_h14 + qwen + moirai-l | 0.6367 | 0.6225 | 0.5937 | 0.0112 | 0.0522 | 0.0930 | 0.0436 |
| vit_h14 + qwen + timesfm-l | 0.7295 | 0.6288 | 0.6315 | 0.0315 | 0.1094 | 0.1779 | 0.0826 |
| dinov2 + qwen + moment-l | 0.7112 | 0.6009 | 0.6026 | 0.0225 | 0.0865 | 0.1471 | 0.0668 |
| siglip + qwen + chronos-l | 0.7498 | 0.6082 | 0.7134 | 0.0695 | 0.1984 | 0.2865 | 0.1421 |
| siglip + e5 + moirai-l | 0.7694 | 0.6781 | 0.7337 | 0.0419 | 0.1435 | 0.2299 | 0.1064 |
| siglip + e5 + moment-l | 0.7562 | 0.6362 | 0.7150 | 0.0565 | 0.1784 | 0.2699 | 0.1277 |
| vit_h14 + e5 + chronos-l | 0.7448 | 0.6403 | 0.6492 | 0.0423 | 0.1364 | 0.2114 | 0.1003 |
| dinov2 + e5 + timesfm-l | 0.7287 | 0.6547 | 0.6322 | 0.0328 | 0.1183 | 0.1916 | 0.0877 |
| *Upper Bound Models* | | | | | | | |
| dino_7b + qwen_8b + moment-l | 0.7959 | 0.6229 | 0.6912 | 0.0619 | 0.1835 | 0.2712 | 0.1323 |
| eva_8b + qwen_8b + chronos-l | 0.7065 | 0.6251 | 0.6891 | 0.0340 | 0.1228 | 0.1943 | 0.0896 |
| dino_7b + mistral_12b + chronos-l | 0.7978 | 0.6491 | 0.7417 | 0.0805 | 0.2172 | 0.3112 | 0.1571 |
| eva_8b + mistral_12b + moirai-l | 0.6464 | 0.7080 | 0.6973 | 0.0189 | 0.0763 | 0.1351 | 0.0625 |
| eva_8b + mistral_12b + timesfm-l | 0.7215 | 0.6795 | 0.7330 | 0.0357 | 0.1267 | 0.2073 | 0.0945 |
| eva_18b + qwen_8b + moirai-l | 0.6619 | 0.6288 | 0.6652 | 0.0161 | 0.0670 | 0.1175 | 0.0548 |
| eva_18b + mistral_12b + moment-l | 0.7123 | 0.6631 | 0.7275 | 0.0335 | 0.1222 | 0.1963 | 0.0902 |
| dino_7b + gemma_27b + moirai-l | 0.7605 | 0.7432 | 0.7581 | 0.0400 | 0.1455 | 0.2393 | 0.1077 |
| eva_8b + gemma_27b + moment-l | 0.7003 | 0.6848 | 0.7183 | 0.0332 | 0.1259 | 0.2067 | 0.0923 |
| eva_18b + gemma_27b + chronos-l | 0.7116 | 0.6866 | 0.7318 | 0.0485 | 0.1579 | 0.2429 | 0.1139 |

*Table 15.* Geometric analysis across modality pairs for all model configurations.

| Model Configuration | Procrustes | | | CKA | | | Mutual KNN | | |
|---|---|---|---|---|---|---|---|---|---|
| | TS–IMG | TS–TXT | IMG–TXT | TS–IMG | TS–TXT | IMG–TXT | TS–IMG | TS–TXT | IMG–TXT |
| *Lower Bound Models* | | | | | | | | | |
| dinov2-b + t5_small + moirai-b | 0.7683 | 0.6343 | 0.7172 | 0.6502 | 0.6467 | 0.5926 | 0.0415 | 0.0773 | 0.0479 |
| vit-b + t5_small + moment-b | 0.5516 | 0.7604 | 0.6702 | 0.7326 | 0.5554 | 0.6261 | 0.1017 | 0.0664 | 0.0711 |
| dinov2-b + t5_small + chronos-b | 0.4965 | 0.6825 | 0.6451 | 0.7499 | 0.5826 | 0.6440 | 0.0969 | 0.0787 | 0.0643 |
| dinov2-b + e5-base + timesfm-b | 0.6577 | 0.7648 | 0.6605 | 0.6339 | 0.5423 | 0.6385 | 0.0753 | 0.0632 | 0.0664 |
| vit-b + e5-b + chronos-b | 0.4769 | 0.7277 | 0.6666 | 0.7662 | 0.5429 | 0.6322 | 0.1065 | 0.0723 | 0.0659 |
| siglip2-b + e5-b + moirai-b | 0.5234 | 0.7520 | 0.4409 | 0.7580 | 0.5650 | 0.7138 | 0.1064 | 0.0645 | 0.1145 |
| siglip2-b + e5-b + moment-b | 0.5444 | 0.7546 | 0.4589 | 0.7161 | 0.5329 | 0.7112 | 0.1189 | 0.0651 | 0.1125 |
| siglip2-b + t5_small + timesfm-b | 0.6345 | 0.7743 | 0.4500 | 0.6569 | 0.5535 | 0.7249 | 0.0953 | 0.0692 | 0.1276 |
| vit-b + qwen_06b + moirai-b | 0.6852 | 0.8092 | 0.7750 | 0.6970 | 0.5505 | 0.5262 | 0.0549 | 0.0652 | 0.0607 |
| dinov2-b + qwen_06b + moment-b | 0.5437 | 0.8066 | 0.7546 | 0.7143 | 0.5236 | 0.5368 | 0.0900 | 0.0713 | 0.0616 |
| vit-b + qwen_06b + timesfm-b | 0.6321 | 0.8224 | 0.7751 | 0.6539 | 0.5248 | 0.5429 | 0.0877 | 0.0741 | 0.0625 |
| siglip2-b + qwen_06b + chronos-b | 0.4578 | 0.7499 | 0.5104 | 0.7625 | 0.5224 | 0.6684 | 0.1448 | 0.0773 | 0.1312 |
| *Middle Bound Models* | | | | | | | | | |
| vit_h14 + t5 + moment-l | 0.5089 | 0.6087 | 0.5775 | 0.7039 | 0.6345 | 0.6351 | 0.1239 | 0.0948 | 0.0893 |
| siglip + t5 + timesfm-l | 0.3999 | 0.5827 | 0.3822 | 0.7758 | 0.6493 | 0.7418 | 0.1895 | 0.1025 | 0.1780 |
| dinov2 + t5 + moirai-l | 0.7239 | 0.5307 | 0.6553 | 0.6575 | 0.7149 | 0.6271 | 0.0455 | 0.1144 | 0.0643 |
| dinov2 + t5 + chronos-l | 0.4585 | 0.6072 | 0.5800 | 0.7635 | 0.6415 | 0.6710 | 0.1200 | 0.0868 | 0.0815 |
| vit_h14 + qwen + moirai-l | 0.6661 | 0.6765 | 0.6947 | 0.6664 | 0.6232 | 0.5729 | 0.0631 | 0.0813 | 0.0697 |
| vit_h14 + qwen + timesfm-l | 0.4620 | 0.6264 | 0.6185 | 0.7518 | 0.6352 | 0.6367 | 0.1299 | 0.1088 | 0.0905 |
| dinov2 + qwen + moment-l | 0.4982 | 0.6957 | 0.6825 | 0.7053 | 0.5787 | 0.5734 | 0.1179 | 0.0976 | 0.0777 |
| siglip + qwen + chronos-l | 0.3956 | 0.6573 | 0.4491 | 0.7770 | 0.6130 | 0.7070 | 0.1868 | 0.0927 | 0.1639 |
| siglip + e5 + moirai-l | 0.3863 | 0.5625 | 0.4235 | 0.8022 | 0.6747 | 0.7138 | 0.1643 | 0.1220 | 0.1855 |
| siglip + e5 + moment-l | 0.3999 | 0.6309 | 0.4653 | 0.7638 | 0.6311 | 0.6819 | 0.1777 | 0.1053 | 0.1617 |
| vit_h14 + e5 + chronos-l | 0.4320 | 0.6237 | 0.6008 | 0.7679 | 0.6125 | 0.6133 | 0.1352 | 0.0965 | 0.1020 |
| dinov2 + e5 + timesfm-l | 0.4694 | 0.5929 | 0.6338 | 0.7299 | 0.6375 | 0.6064 | 0.1279 | 0.1051 | 0.0881 |
| *Upper Bound Models* | | | | | | | | | |
| dino_7b + qwen_8b + moment-l | 0.3324 | 0.6671 | 0.5317 | 0.7961 | 0.5946 | 0.6362 | 0.2221 | 0.0948 | 0.1413 |
| eva_8b + qwen_8b + chronos-l | 0.4913 | 0.6320 | 0.5152 | 0.7116 | 0.6401 | 0.6714 | 0.1179 | 0.0876 | 0.1160 |
| dino_7b + mistral_12b + chronos-l | 0.3231 | 0.5962 | 0.4191 | 0.8398 | 0.6315 | 0.7154 | 0.2101 | 0.1111 | 0.1737 |
| eva_8b + mistral_12b + moirai-l | 0.6450 | 0.5134 | 0.5138 | 0.6662 | 0.7290 | 0.6641 | 0.0652 | 0.1275 | 0.1168 |
| eva_8b + mistral_12b + timesfm-l | 0.4799 | 0.5401 | 0.4517 | 0.7213 | 0.6707 | 0.7026 | 0.1316 | 0.1164 | 0.1165 |
| eva_18b + qwen_8b + moirai-l | 0.6133 | 0.6675 | 0.5722 | 0.6851 | 0.6258 | 0.6263 | 0.0743 | 0.0840 | 0.1053 |
| eva_18b + mistral_12b + moment-l | 0.4916 | 0.5785 | 0.4613 | 0.7132 | 0.6614 | 0.6949 | 0.1187 | 0.1053 | 0.1180 |
| dino_7b + gemma_27b + moirai-l | 0.4112 | 0.4370 | 0.3874 | 0.7839 | 0.7478 | 0.7347 | 0.1424 | 0.1451 | 0.1804 |
| eva_8b + gemma_27b + moment-l | 0.5140 | 0.5347 | 0.4716 | 0.6896 | 0.6885 | 0.6799 | 0.1160 | 0.1196 | 0.1235 |
| eva_18b + gemma_27b + chronos-l | 0.4803 | 0.5180 | 0.4331 | 0.7178 | 0.7083 | 0.7267 | 0.1316 | 0.1228 | 0.1409 |

