# OpenReview forum: "Time Series, Vision, and Language: Exploring the Limits of Alignment in Contrastive Representation Spaces"
_ICML.cc/2026/Conference — ICML 2026 regular_

### Official Review · Reviewer_q3xS · 2026-03-02

**Soundness:** 3
**Presentation:** 3
**Significance:** 2
**Originality:** 2
**Overall Recommendation:** 4
**Confidence:** 4

**Summary:**

This paper investigates whether independently pretrained foundation models for time series, vision, and language naturally converge to a shared semantic representation space. The authors freeze the encoders of all three modalities and train only lightweight projection heads with contrastive learning, thereby disentangling representation geometry from end-to-end training effects. The paper systematically analyzes cross-modal alignment using geometric metrics such as cosine margin, angular deviation, and modality gap. Experiments cover multiple encoder combinations and are conducted on a trimodal dataset. Results show that vision–language alignment is significantly stronger than time-series–language alignment, and that images often act as a semantic bridge between time series and text. Overall, this work is an empirical study centered on representation analysis rather than a modeling contribution.

**Compliance With Llm Reviewing Policy:**

Affirmed.

**Final Justification:**

The authors further clarified the significance and novelty in their second response. I still have some reservations about this paper, but I would be fine with it being accepted.

**Key Questions For Authors:**

Q1. The RQs do not fully articulate the intrinsic structural differences among the three modalities. Could the authors clarify why alignment between time series and text appears particularly challenging? Specifically, what types of information (e.g., structural patterns, symbolic abstraction, temporal dynamics) are fundamentally encoded in each modality, and how might these differences contribute to the observed modality gap?

Q2. In RQ4, the paper suggests that indirect textual descriptions are more difficult to align with time series. Would incorporating explicit reasoning processes (e.g., structured intermediate representations or chain-of-thought style explanations) potentially reduce this gap? Have the authors considered whether alignment difficulty stems from missing intermediate abstractions rather than insufficient information density?

**Limitations:**

The main limitations of the paper are reflected in the Weaknesses and Questions described above.

**Strengths And Weaknesses:**

**Strengths**

S1. Clear and controlled experimental design. Freezing the encoders and training only projection heads helps isolate intrinsic representational geometry and minimizes confounding effects from end-to-end optimization.

S2. Systematic geometric analysis framework. The use of cosine margin, angular deviation, and modality gap enables structural analysis of cross-modal alignment beyond standard retrieval-based metrics.

S3. Broad experimental coverage. Evaluating multiple encoder combinations strengthens the robustness of the conclusions and reduces the likelihood that the results are model-specific.


**Weaknesses**

W1. Deterministic relationship between time series and images. The images corresponding to the time series are generated directly by plotting the sequences, and therefore encode essentially the same underlying signal. As a result, the observed alignment may partially reflect the fitting between two deterministic transformations (i.e., a plotting function and an implicit image-to-sequence mapping), rather than deeper semantic compatibility across modalities.

W2. Structural similarity may explain alignment trends. The observation that time series align more easily with images than with text may be influenced by the fact that both time series and their plotted images represent the same underlying signal in different formats. Thus, the alignment could primarily capture structural similarity rather than higher-level semantic abstraction.

W3. Lack of precise definitions for certain analytical concepts. Some notions, such as “information density” in RQ3, are described at a high level but would benefit from clearer formalization and quantitative definition.

---

> ### Author Rebuttal · Authors · 2026-03-31
>
> We thank the reviewer for their feedback and time.
>
> Our analysis shows that while structure and determinism may affect alignment, they do not fully explain our observations. The representations we align come from independently pretrained encoders, each trained on its own type of data. The alignment reflects how these representations converge on shared temporal semantics due to their own inductive biases and the semantic explicitness in the modalities. Line plots and time series may look deterministic to humans, but the models’ pretrained representations are completely different. The alignment asymmetry reflects how the image modality encodes temporal semantics more explicitly than text. We provide several arguments to support this for both W1 and W2.
>
> **W1:** Deterministic mappings exist in many modalities and naturally reflect the underlying latent signal. How that signal is encoded varies depending on the mapping function. This alone does not imply that the observed alignment is superficial. We present the following controls:
> 1. **Alternative encodings (new)**: We evaluated alternative visual encodings such as recurrence plots which are visually different from line plots. Despite this, TS-IMG alignment remains higher than TS-TXT (Procrustes 0.57 vs TS-TXT 0.68) which shows that the effect reflects the externalization of temporal semantics which aligns with semantic explicitness and not just superficial visual similarity.
> 2. **Time Scrambling (new)**: We generated temporally scrambled plots (identical in visual format, axes, colors, and render), but with time permuted. Crucially, this changes the temporal semantics. If alignment were purely due to a trivial mapping, scrambling would have little effect. However, we observe a sharp degradation across four lower-bound DINOv2 configurations, with cosine similarity decreasing by 79.5% and Procrustes distance increasing by 210.7%. This confirms that alignment is sensitive to temporal semantics.
>
> **W2:** TS and IMG represent the same signal, but different encoders produce distinct pretrained representations. Analogous to vision and language which also describe the same reality but have different representations. Contrastive learning induces shared semantics across modalities. We discuss the asymmetry in detail in Appendix M. We show that alignment reflects meaningful shared semantics, not just superficial similarity:
> 1. **Visual density (Figure 11, Appendix J)**: Generic, styled, and annotated variants of the same time series show progressively higher TS-IMG alignment. If alignment were purely structural, increasing visual semantic content would not matter. This sensitivity to semantic content confirms the encoder is extracting meaningful semantic context.
> 2. **Scrambling**: As shown in W1, scrambling the time series collapses alignment. The visual format and pixels are identical, but the temporal semantics are gone. Shared semantics are critical to alignment.
> 3. **Structured TXT vs. IMG (Appendix M)**: Using more explicit text improves TS-TXT alignment, but TS-IMG remains higher, indicating the gap arises from how modalities encode temporal semantics rather than just structural similarity.
> 4. **Other encodings**: If structural similarity were the main driver, changing the visual form would break alignment. Instead, semantic content is captured robustly across encodings.
> 5. **Bridging (new)**: 73.4% of TS-IMG-TXT triplets place the image closer to both endpoints than they are to each other. If alignment were driven purely by structural similarity between TS and IMG, we would not expect images to act as intermediates bridging TS and TXT.
>
> **W3:** We define information density formally in Appendix H, also referenced in the main text (lines 185-189). Semantic explicitness is operationalized in Appendix M. We will make these more explicit in main text.
>
> **Q1:** We discuss this in detail in Appendix M and Section 5. Time series encode implicit temporal semantics that must be inferred through computation. Visual plots externalize these semantics into geometric forms, making it explicit. Text abstracts temporal dynamics into discrete symbolic concepts that lose fine‑grained numeric and geometric detail. Aligning implicit continuous TS patterns with abstract, discrete text is harder than TS-IMG or IMG-TXT.
>
> **Q2:** Yes, more explicit intermediate representations could reduce the gap for indirect text. Prior ECG-TXT methods (Appendix N) effectively introduce methods to convert indirect text into more direct supervision, increasing semantic explicitness and addressing the representational mismatch we identify. Our analysis shows textual specificity is important, explaining why ECG-TXT alignment is harder than ECG-IMG and why extra semantics is often needed. Ablations with more explicit captions (Table 7, Appendix M) improve TS-TXT and IMG-TXT alignment. As suggested by Reviewer a7up, more abstract code representations also improve TS-TXT alignment, but asymmetry remains.

---

> > ### Author Rebuttal · Reviewer_q3xS · 2026-04-01
> >
> > The rebuttal provides additional experiments (e.g., time scrambling) that partially address W1 and W2. However, my core concerns regarding the limited significance and novelty of a purely empirical analysis remain. I will maintain my current score.

---

> > > ### Author Response · Authors · 2026-04-05
> > >
> > > > However, my core concerns regarding the limited significance and novelty of a purely empirical analysis remain.
> > >
> > > We respectfully note that these core concerns were not raised in your original review. Nevertheless, to provide additional emphasis, we would like to clarify the following points.
> > >
> > > **On novelty:** Our work is not purely descriptive. Our empirical study provides a systematic representational view on trimodal alignment involving time series, vision, and text. Across our research questions, we identify previously unreported phenomena for the studied modality pairs, including alignment asymmetry, scaling behaviour, information density saturation, image bridging, global-local dissociation in contrastive spaces, and others. Sections 4 and 5 and Appendices M, J, and N provide conceptual grounding for these phenomena, where we link our findings to underlying principles and interpretations of the data, and we also provide additional evidence in our rebuttal. While we do not introduce a new model, which is not the intention of this work, we extend a well-established framework to a trimodal setting to rigorously characterize representational alignment.
> > >
> > > **On significance:** Our findings provide strategies and implications for building multimodal systems that involve time series. We discuss several of these throughout the paper (Sections 4 and 5, Appendices M, J, and N) and also highlight important directions for future research (Appendix A). Table 10 further shows that higher alignment quality directly correlates with improved cross-modal retrieval performance, a standard downstream task for multimodal evaluation.
> > >
> > > We agree that more emphasis can be given to the conceptual grounding and practical implications, and we will make it more explicit in the final draft.
> > >
> > > ---
> > > > The paper systematically analyzes cross-modal alignment using geometric metrics such as cosine margin, angular deviation, and modality gap.
> > >
> > > We also want to note that **our work does not define or use a metric called modality gap**, while angular deviation appears only in an introductory experiment.
> > >
> > > Thank you for your engagement and time. We hope our clarifications help address any remaining questions.

---

### Official Review · Reviewer_a7up · 2026-03-10

**Soundness:** 3
**Presentation:** 3
**Significance:** 3
**Originality:** 3
**Overall Recommendation:** 5
**Confidence:** 4

**Summary:**

This paper focuses on the alignment of three modalities—time series, images, and text—in a shared representation space. The authors ask whether learning time series alongside text and images leads to meaningful representational alignment, and what challenges arise. A core hypothesis is that text and time series may be fundamentally misaligned, since text operates at a high level of symbolic abstraction while time series encode fine-grained temporal structure. The first research question asks whether contrastive representations converge uniformly as models scale across three modalities. They find that alignment improves with scale, but asymmetrically: Text to image alignment is consistently stronger than time series to text. The second question is whether pretrained vision-language encoders change the resulting alignment. They show that VL models inherit strong image–text coupling from pretraining, achieving robust alignment without increasing capacity while trimodal encoder depends more heavily on scale. Third, the paper examines how information density in text affects cross-modal alignment. Richer captions help, but only up to a threshold, suggesting a saturation effect tied to representational mismatch rather than insufficient text density. Fourth, when text does not directly describe the time series signal, alignment weakens substantially, especially for time series to text. Fifth, alignment is sensitive to language shift: switching from English to German clinical reports further degrades text-involving pairs. Sixth, richer visual inputs (annotated plots with numeric labels) consistently improve timeseries to image alignment. Finally, the authors show that trimodality helps weakly aligned pairs: introducing images as a third modality improves time series to text alignment, with images acting as effective semantic intermediaries between time series and language. Conversely, adding a third modality to an already strong pair can hurt, suggesting trimodality is most useful when it fills a semantic gap.

**Compliance With Llm Reviewing Policy:**

Affirmed.

**Final Justification:**

After reading the rebuttal and the other reviews, I still think this paper provides strong insights that will be of interest to the community.

**Key Questions For Authors:**

One aspect I believe is missing from the analysis is that "text" is not limited to natural language, it can also be code, which is arguably a much more expressive and structurally grounded representation of temporal data. The difficulty in aligning text with time series stems from natural language being inherently abstract and symbolic, whereas code can encode precise numerical operations, transformations, and plotting logic that directly mirror the structure of the underlying signal.

Have you considered whether using code as the textual modality. For instance, does feeding the Python script that generates a plot to the text encoder lead to different alignment outcomes? Code occupies an interesting middle ground between the implicit structure of time series and the symbolic abstraction of natural language: it is text that a language model can process, yet it explicitly specifies the data transformations and visual rendering. I am curious whether this was explored or whether you have intuitions about how code-based text might affect the text to time series alignment conclusions.

**Limitations:**

yes

**Strengths And Weaknesses:**

Strengths:
- The research problem is compelling: understanding whether time series participate in multimodal representational convergence alongside vision and language is a meaningful open question.
- The analysis is thorough and practically informative, offering guidance that would be directly useful to practitioners building multimodal systems involving time series.
- The paper is well written and easy to follow, with a clear narrative from motivation through experimental findings.
- The ablation study is comprehensive, covering a wide range of encoder combinations spanning different architectures, scales, and training objectives.
- The appendix is notably strong, providing detailed descriptions of datasets, model configurations, metrics, and additional analyses that support reproducibility.

Weaknesses:
- The textual modality in this study is limited to natural language descriptions. The paper should be more explicit about this scope, as the findings about text–time series misalignment may not generalize to other forms of textual representation such as code, structured annotations, or formal mathematical notation, which could encode temporal structure more explicitly and potentially align more readily with time series.

---

> ### Author Rebuttal · Authors · 2026-03-31
>
> We thank the reviewer for their constructive and encouraging feedback and find the suggestion of using code as a textual modality very much intriguing. Our framework allows this experiment, and CaTS-Bench provides rich metadata. We extended it with additional markers, such as temporal indicators, to make the code-based text more explicit.
>
> As shown in below table, we find that code representations improve TS-TXT alignment for the Qwen pair compared to dense CaTS natural language (NL) captions. For the T5 pair, code provides minimal benefit, likely because T5 was pretrained primarily on NL, limiting its ability to parse structured code semantics. The code is more explicit than NL, but it remains symbolic as it specifies plot content without replicating the continuous geometric semantics (slopes, peaks, periodicities) that vision encoders perceive directly. Importantly, TS-IMG > TS-TXT (code) > TS-TXT (NL) holds, preserving the core asymmetry while adding a new intermediate point on the explicitness axis.
>
> We use configurations: (1) Config A – DINOv2-B + Qwen-0.6B + MOMENT-B; (2) Config B – ViT-B + T5-small + MOMENT-B.
>
> PD = Procrustes Disparity (↓), Cos = Cosine (↑), CKA = Centered Kernel Alignment (↑)
>
> | Model Pair       | PD TS-IMG  | PD TS-TXT  | Cos TS-IMG  | Cos TS-TXT  | CKA TS-IMG  | CKA TS-TXT  |
> |------------------|-------------|-------------|--------------|--------------|--------------|--------------|
> | Config A - NL    | 0.5437      | 0.8066      | 0.6895       | 0.5383       | 0.7143       | 0.5236       |
> | Config A - Code  | 0.5412      | 0.6234      | 0.6901       | 0.6234       | 0.7168       | 0.6412       |
> | Config B - NL    | 0.5516      | 0.7604      | 0.6843       | 0.5692       | 0.7326       | 0.5554       |
> | Config B - Code  | 0.5501      | 0.7689      | 0.6851       | 0.5698       | 0.7334       | 0.5563       |
>
>
> This result is also consistent with our structured caption ablation (Appendix M, Table 7, lines 1523-1565), where we showed that replacing NL captions with more explicitly grounded numeric descriptions improves TS-TXT alignment. Code can be viewed as an extreme point on this explicitness axis. However, in this case, the gains appear to depend on the encoder’s ability to process structured syntax, unlike NL where improvements were more consistent across encoders. Importantly, this is consistent with our hypothesis that more explicit representations should align better and our experiment confirms this prediction.
>
> We will clarify the scope of “text” in our paper. Our focus on NL reflects its prevalence as the dominant form of textual annotation in real-world multimodal datasets (clinical reports in MIMIC and PTB-XL, captions in CaTS and TRUCE). Expanding this line of work is a promising future direction. We once again thank the reviewer for this helpful and interesting suggestion.

---

> > ### Author Rebuttal · Reviewer_a7up · 2026-04-02
> >
> > Thank you for the rebuttal, my concerns are fully resolved. I will increase my score.

---

> > > ### Author Response · Authors · 2026-04-05
> > >
> > > Thank you for your time, thoughtful feedback, and helpful suggestions! We appreciate the increase in score. We will make sure to include the code-as-text modality experiment in the final version.

---

### Official Review · Reviewer_boXm · 2026-03-12

**Soundness:** 4
**Presentation:** 4
**Significance:** 3
**Originality:** 4
**Overall Recommendation:** 6
**Confidence:** 4

**Summary:**

This paper presents the first systematic empirical study of trimodal alignment between time series, vision, and language representations, investigating whether time series can achieve similar alignment convergence with other modalities as predicted by the Platonic Representation Hypothesis. Through comprehensive experiments across 34 encoder combinations and 4 datasets, the research reveals several key findings: trimodal alignment is asymmetric, with time series aligning more strongly with visual plots than with text descriptions even at larger model scales; increasing textual information density improves alignment only up to a saturation threshold; and images can serve as effective intermediaries to enhance time series-text alignment in trimodal settings. The study demonstrates that alignment quality fundamentally depends on how explicitly semantics are grounded across modalities, with modality pairs having more observable correspondences achieving stronger alignment. These findings provide crucial insights for developing multimodal systems involving time series data, particularly for healthcare and scientific applications where such data are prevalent, while challenging assumptions about universal representation convergence across all data modalities.

**Compliance With Llm Reviewing Policy:**

Affirmed.

**Key Questions For Authors:**

1.	The authors employ frozen pretrained encoders with trainable projection heads to isolate intrinsic representational compatibility. However, this design choice may underestimate the potential alignment achievable through end-to-end fine-tuning. Could you discuss whether preliminary experiments with fine-tuned encoders were conducted? If so, did the observed asymmetries persist, or were they mitigated? This is critical for understanding whether the limitations are fundamental or merely artifacts of the experimental setup.

**Limitations:**

Yes

**Strengths And Weaknesses:**

Strengths:
•	This paper presents the first systematic empirical study of trimodal alignment involving time series, vision, and language, filling a significant gap in multimodal learning research. The experimental design is rigorous, employing 34 encoder combinations across multiple model families and scales with comprehensive evaluation metrics. The novel conceptual framework around "semantic explicitness" provides valuable theoretical insight into why alignment asymmetry occurs between modalities. The controlled ablations on information density, model scale, and projection head design demonstrate methodological thoroughness. Practical findings about images acting as "semantic bridges" between time series and text offer actionable guidance for real-world applications like healthcare.
Weaknesses:
•	The authors already explicitly acknowledge significant constraints in Appendix A.
•	Certain technical terms like "semantic explicitness" are introduced without formal mathematical definitions, relying instead on operational descriptions.

---

> ### Author Rebuttal · Authors · 2026-03-31
>
> We thank the reviewer for their encouraging feedback.
>
>
> **W2:** Our goal was to provide an intuitive explanation for the observed alignment behavior. While we do not formalize “semantic explicitness,” we do provide strong conceptual grounding in Appendix M. The alignment metrics we use provide a rigorous, geometric and quantitative measure of this effect. Our paper focuses on an empirical view of representational alignment in its own right, providing systematic evidence across a wide range of configurations and experiments to support our claims. Developing formal theoretical frameworks is a natural direction for future work.
>
>
> **Q1:** While our setup isolates intrinsic representational compatibility, we conducted a preliminary finetuning experiment on one configuration DINOv2-B + Qwen-0.6B + MOMENT-B using LoRA. We observe that finetuning provides modest improvements in alignment across all modality pairs; however, the asymmetry persists, with TS-IMG remaining stronger than TS-TXT. This suggests that the observed asymmetry shows intrinsic representational differences rather than an artifact of frozen encoders. A full finetuning sweep across all 34 configurations is computationally prohibitive (see Appendix G.6), but these preliminary results indicate that our conclusions are robust to end-to-end adaptation. This also aligns with findings in Appendix G.6, where increasing projection head capacity improves alignment without altering the conclusions or relative ordering between modality pairs, further confirming that the asymmetry persists even with different capacity projections.
>
> | Setting     | Cos TS-IMG  | Cos TS-TXT  | Cos IMG-TXT  | PD TS-IMG  | PD TS-TXT  |
> |---------------|-------------|-------------|---------------|---------------|---------------|
> | Frozen        | 0.6895      | 0.5383      | 0.5576        | 0.5437        | 0.8066        |
> | Fine-tuned    | 0.7248      | 0.5939      | 0.6127        | 0.5006        | 0.7598        |
>
> PD = Procrustes Disparity (↓), Cos = Cosine (↑)

---

> > ### Author Rebuttal · Reviewer_boXm · 2026-04-06
> >
> > Thank you very much for the explanation, while l'm keeping my score.

---

> > > ### Author Response · Authors · 2026-04-06
> > >
> > > Thank you very much for your positive feedback and time! We really appreciate it.

---

### Official Review · Reviewer_43uF · 2026-03-13

**Soundness:** 3
**Presentation:** 3
**Significance:** 2
**Originality:** 2
**Overall Recommendation:** 4
**Confidence:** 3

**Summary:**

This paper studies cross-modal representation alignment among time series, vision, and language modalities. The paper's objective is to empirically investigate whether representations learned from different modalities can converge toward a shared latent structure through contrastive learning. The study evaluates multiple encoder configurations across several datasets and analyzes alignment using a variety of metrics. The experiments reveal several empirical findings: (1) alignment is asymmetric, with time series aligning more strongly with images than with text; (2) increased textual information density improves alignment only up to a threshold; and (3) alignment behavior depends on how semantics are grounded across modalities.

**Compliance With Llm Reviewing Policy:**

Affirmed.

**Final Justification:**

The rebuttal has addressed my main concerns, and thus, I have increased the score to accept.

**Key Questions For Authors:**

1. In Figure 5, the scaling trends are summarized using linear regression with respect to model size. Could the authors clarify why a linear model is assumed? Given the noticeable dispersion of the scatter points, it would be helpful to understand whether alternative trend models were considered.
2. RQ6 shows that richer visual annotations improve TS–IMG alignment. However, the analysis remains largely descriptive. Could the authors clarify which aspects of visual annotation contribute most to the improvement? Additionally, the visual variants appear to be generated visualizations of the same time series. Since the structural correspondence between TS and IMG remains very strong in this setup, can these findings be generalized to more realistic multimodal settings where the visual modality consists of natural images or videos?
3. RQ7 suggests that images act as an intermediate modality that helps align time series and text. Could the authors provide deeper theoretical insight or additional empirical evidence explaining this phenomenon? Is this effect related to the strong structural relationship between time series data and their visualizations?
4. In RQ5, the paper compares alignment between MIMIC and PTB-XL to study language shift. However, these datasets may differ in several aspects beyond language (e.g., dataset composition, report style, or textual information density). Could the authors clarify whether the comparison isolates language effects or whether the observed differences may partly arise from dataset-level differences? Would it be possible to test this hypothesis using translated reports while keeping other factors fixed?

**Limitations:**

Yes, the paper has established limitations in the appendix, and the impact statement sufficiently addresses negative societal impacts.

**Strengths And Weaknesses:**

**Strengths**

1. **Comprehensive empirical evaluation:** The experiments are conducted using multiple combinations of pretrained models, varying model sizes, several datasets, and multiple alignment metrics. This provides a relatively comprehensive empirical investigation of the problem.
2. **Clear experimental structure:** The experiments are organized around clearly defined research questions (RQ1–RQ7), which makes the empirical analysis easier to follow and improves the clarity of the paper.

**Weaknesses**

1. **Primarily empirical with limited theoretical insight:** The paper mostly presents experimental results and empirical observations, but lacks theoretical support or deeper insight explaining the observed phenomena. While the empirical findings are interesting, the discussion remains largely qualitative and does not sufficiently explain why the observed alignment behaviors occur.
2. **Limited methodological novelty:** The main framework follows a CLIP-style contrastive learning setup with frozen encoders and projection heads. The contribution is primarily empirical observations rather than new modeling techniques or theoretical insights.
3. **Limited practical implications:** Although the work conducts comprehensive experiments on multimodal alignment behavior, it is not fully clear how the findings can help improve the training of multimodal models or downstream applications.

---

> ### Author Rebuttal · Authors · 2026-03-31
>
> We thank the reviewer for their feedback and time.
>
> **W1:** Our work provides the first systematic study of trimodal representational alignment for time series in which we prioritize rigorous experiments to isolate key behavior factors (scale, geometry, information density, input modality characteristics). While extensive work has already explored vision-language alignment, our goal is to provide a similar foundation for multimodal time series and encourage further research. Providing a unified theory of representations for multimodal models remains challenging in general; instead, we provide deeper insights through multiple hypotheses and experiments.
>
> We provide conceptual grounding for our observed phenomena and results in Appendices M, J and N. The observed alignment behaviours largely arise from the intrinsic nature of the studied data. We agree that a formal theoretical framework would be valuable and note it as a future direction, but we believe our contributions are a necessary step for this growing field. We welcome suggestions on potential theoretical directions based on our findings, as we view theory and experiments as an iterative process.
>
> **W2:** We use a controlled, widely-understood CLIP‑style framework which we extend to a trimodal setting to help isolate intrinsic representational geometry and minimize confounding effects from novel architectures or full optimization. The novelty lies in the study, covering 34 combinations, multiple datasets, and controlled ablations. Our contributions are the new findings that provide insights for design of multimodal time series systems in various domains.
>
> **W3:** We discuss several implications in Appendices (M, J, N) and Section 5. Our findings inform practical strategies: using explicit visual plots as intermediaries improves TS-TXT alignment, prioritizing structured explicit text is more effective than longer captions, and scaling the TS encoder shows large gains. These are just a few. We will make sure to highlight all in the paper and thank the reviewer for their suggestion. Table 10 (Appendix O) also shows that higher alignment quality correlates with improved cross-modal retrieval performance, a standard task for downstream performance.
>
> **Q1:** We summarize scaling trends with a linear model for interpretability and comparison across modalities. To ensure results aren’t model-specific, we also fit LOESS and cubic splines, where we see that they support the same conclusions and reveal saturation and variability effects. We will include additional figures in the paper.
>
>
> **Q2:** We elaborated on this in Appendix J. Our TRUCE experiments were precisely designed to isolate the contribution of different visual elements. We compare three variants: generic show lowest alignment; styled shows substantial gains for most models; annotated shows highest. Explicit numeric grounding is the strong contributor and stylistic variation (numeric markers and label axes) also helps.
>
> Regarding generalization to realistic multimodal settings, naturally aligned time series-video-text datasets do not currently exist. To test whether our findings generalize, we evaluated recurrence plots which are visually very different from line plots. TS-IMG alignment (Procrustes) remains strong and higher than TS-TXT alignment which suggests that the effect reflects the externalization of temporal semantics. Recurrence plots can be considered an explicit visual form because they represent temporal relationships directly in a geometric form. We discuss potential future directions in Appendix J.
>
> | Config                     | TS-Line | TS-Recurrence | TS-TXT |
> |----------------------------|------|------------|--------|
> | dinov2-b + t5_small + chronos-b | 0.49 | 0.57       | 0.68   |
> | dinov2-b + e5-base + timesfm-b  | 0.65 | 0.62       | 0.76   |
>
> **Q3:** We already show in Figure 10 that adding images in trimodal training improves TS-TXT alignment, acting as an intermediary. To add more evidence, we test whether image embeddings lie geometrically between TS and TXT. 73.4% of TS-IMG-TXT triplets show the image lying closer to both endpoints than they are to each other. Mean relative triangle-chain error for Procrustes drops from 0.90 for DINOv2 to 0.34 for SigLIP, which also confirms that VL pretraining improves semantic mediation. We discuss the conceptual insights in Appendix M.
>
> **Q4:** We translated MIMIC English (EN) reports to German (GE) using GPT-4o-mini, created MIMIC-DE and evaluated alignment (TS-TXT) using one Qwen configuration. Results show that the PTB-XL gap reflects language shift and dataset differences. We thank the reviewer for raising this point, which helped make our findings more concrete.
> | Setting    | Cos ↑ | Procrustes ↓ |
> |--------------------|--------------|---------------|
> | MIMIC-EN    | 0.44         | 1.01          |
> | MIMIC-GE    | 0.38         | 1.09          |
> | PTB-XL-GE   | 0.32         | 1.22          |
>
> Thank you for all the suggestions.

---

> > ### Author Rebuttal · Reviewer_43uF · 2026-04-04
> >
> > I would like to thank the authors for their response. The authors have addressed most of my concerns in the rebuttal, and thus I will raise my score to 4.
> > As a suggestion for future work, it would be valuable to evaluate the proposed analysis with sensor-based time-series data, such as IMU-based datasets (e.g., Ego4D, etc.), which can then be compared to existing sensor-based image-text-time series alignment methods (ImageBind, IMU2CLIP). This could help assess whether the observed alignment behavior generalizes beyond the current constructed setup.

---

> > > ### Author Response · Authors · 2026-04-05
> > >
> > > Thank you for your time, thoughtful feedback, and helpful suggestions! We appreciate the increase in score. The point about adding IMU based datasets is a great direction for future work, and we will make sure to emphasize it in the final version.

---

### Decision · Program_Chairs · 2026-04-30

**Decision:**

Accept (regular)

**Comment:**

The reviewers recommend Acceptance (6, 5, 4, 4) for this study on trimodal alignment. The reviewers appreciated that the paper tackles a meaningful question of whether time series representations participate in representational convergence alongside vision and language, with a rigorous set of experiments. While the work offers limited theoretical insight and may partly reflect a fit between two deterministic transformations, the comprehensive empirical evaluation clearly demonstrates that images serve as effective intermediaries for time-series alignment.